# Temporal dynamics of collateral RNA cleavage by LbuCas13a in human cells
Jorik Frederik Bot [1,2,4], Zhihan Zhao [1,2,3,4], Mengyuan Li [1,2], Darnell Kammeron [1,2], Peng Shang [1,2] & Niels Geijsen [1,2]

CRISPR-Cas13 exclusively targets RNA. In prokaryotes, Cas13 cleaves both target and non-target RNA indiscriminately upon activation by a specific target RNA, but in eukaryotic cells collateral cleavage activity has been limited. Here we report that LbuCas13a exhibits strong collateral RNA cleavage activity in human cells when delivered as ribonucleoprotein, independent of cell line and targeting both exogenous and endogenous transcripts. Collateral RNA cleavage starts within 50 minutes of ribonucleoprotein delivery resulting in major alterations to the total RNA profile. In response to the collateral RNA cleavage, cells upregulate genes associated with the stress and innate immune response, ultimately leading to apoptotic cell death. This enables us to use LbuCas13a as a flexible and repeatable target-RNA-specific cell elimination tool. Finally, using both total RNA sequencing and Nanopore sequencing, we find that LbuCas13a activation leads to rapid and near-global depletion of cytoplasmic RNAs, and that cleavage occurs at specific nucleotide positions.

CRISPR-Cas is a prokaryotic adaptive immune system that protects bacteria and archaea from invading viruses and plasmids[1–3]. Since its initial discovery, an impressive diversity of CRISPR-Cas systems has been revealed[4]. Especially the Class 2 DNA targeting effector proteins, Cas9 and Cas12, have been extensively studied and have found widespread adoption for biotechnological and clinical use[5,6]. A recently discovered Class 2 system, CRISPR-Cas13, targets and cleaves RNA. The signature element of CRISPR-Cas13 systems are the two Higher Eukaryotes and Prokaryotes Nucleotide-binding (HEPN) RNase domains of the effector protein[7,8]. When a Cas13 protein is activated by a guide-matching target RNA, these domains cooperate to form a single active RNA nuclease site[9–15]. This catalytic site is located on the outside of the protein, facing away from the guide RNA-target RNA complex[16–20]. As a consequence, Cas13 cleaves both target and non-target RNA indiscriminately in vitro[9–13,15,21] and in bacteria[9,10,12,22] when activated by a specific target RNA. In vitro, this collateral RNA cleavage has been employed to develop portable, rapid, and highly sensitive nucleic acid detection methods[21,23–25], including for Coronavirus disease 2019 (COVID-19) diagnostics[26–28]. In bacteria, collateral RNA cleavage probably serves to protect bacterial communities by inducing dormancy after phage infection[22,29].

Initially, no evidence of collateral cleavage was found when Cas13 was first tested in eukaryotic cells[11,14,30]. Instead, Cas13 induced efficient and specific target RNA knockdown. Because Cas13-based knockdown was

shown to have higher specificity and efficiency than RNAi[11,14,30–34] and CRISPRi[11,34], it has seen enthusiastic adoption as a specific RNA knockdown tool. For example, Cas13 has been adapted for use in mice[31,35–37], fish[35,38], yeast[39], fly[34], plants[40,41], and as an anti-viral tool[42–44]. However, recently several groups have reported that Cas13 is capable of collateral cleavage in eukaryotes in certain conditions, in some cases causing cytotoxicity[13,45–51]. To further advance CRISPR-Cas13 as a specific RNA knockdown method, some groups have made progress to minimize the Cas13 collateral cleavage effects[49,52]. We took the opposite approach and set out to exploit Cas13's collateral cleavage activity as a target-RNA-specific cell elimination tool. A comparison of the collateral cleavage activity of different Cas13 orthologs identified LbuCas13a as a particularly active Cas13 variant. We set out to explore the on-target and collateral RNA cleavage activity of LbuCas13a in cells. We observed that specifically ribonucleoprotein (RNP) delivery of LbuCas13a induces robust collateral RNA cleavage in human cells. The collateral activity is cell line and RNP delivery method independent, and is induced by targeting both exogenous and endogenous transcripts. The resulting collateral RNA cleavage leads to transcriptomic changes associated with the activation of an innate immune response. We demonstrate that collateral RNA cleavage triggers target RNA-specific apoptosis, which is strongly correlated with the expression level of the target RNA. Consequently, LbuCas13a can be used as a target RNA-specific cell selection tool which eliminates specific cell types from a heterogenous cell population. We

¹Dept. of Anatomy & Embryology, Leiden University Medical Center, Leiden, The Netherlands. ²The Novo Nordisk Foundation Center for Stem Cell Medicine (reNEW), Leiden node, Leiden, The Netherlands. ³Biomedical Pioneering Innovation Center, Peking-Tsinghua Center for Life Sciences, Peking University Genome Editing Research Center, State Key Laboratory of Gene Function and Modulation Research, School of Life Sciences, Peking University, Beijing, China. ⁴These authors contributed equally: Jorik Frederik Bot, Zhihan Zhao. ✉e-mail: p.shang@lumc.nl; n.geijsen@lumc.nl

demonstrate that this can be used to enrich for cells that have undergone gene editing, and select against cancer cells by targeting an overexpressed oncogene. Finally, using both total RNA sequencing with ERCC spike-in RNAs and Nanopore sequencing, we observed that LbuCas13a collateral RNA cleavage leads to near-global depletion of cytoplasmic RNAs. This cleavage occurs at very specific positions, often within the loops of stem-loop structures. In conclusion, here we describe the highly active LbuCas13a, the temporal dynamics of target and collateral RNA cleavage, the cellular response to collateral RNA cleavage, the application of LbuCas13a as a negative cell selection tool, and the identity and cleavage positions of the RNAs subject to collateral RNA cleavage by LbuCas13a.

## Results

### LbuCas13a exhibits strong collateral RNA cleavage activity in human cells

The enzymatic activity of different Cas13 orthologs in vitro is known to range over seven orders of magnitude[21]. We hypothesized that the Cas13 orthologs with the strongest collateral cleavage activity in vitro are the most likely to show any collateral cleavage in human cells. Therefore, we compared the collateral RNA cleavage activity of six different CRISPR-Cas13 effectors (LbuCas13a, LwaCas13a, BzCas13b, PspCas13b, RspCas13d, and RfxCas13d) in vitro, as well as in cells. We produced and purified all six recombinant proteins (Supplementary Fig. 1a). Next, the in vitro collateral RNA cleavage activity of these proteins was assessed using the RNaseAlert® assay (IDT), which quantitatively detects non-specific RNA cleavage. LbuCas13a exhibited the strongest in vitro collateral cleavage activity (Supplementary Fig. 1b). RfxCas13d did not show any activity in this assay, nor did it show any activity using two other guide RNAs (Supplementary Fig. 1c), suggesting we did not manage to produce functional recombinant RfxCas13d protein. It was therefore excluded it from further testing. To test these orthologs in human cells, we transfected them as recombinant protein together with guide RNAs into HAP1 cells constitutively overexpressing a fluorescence-inactivated EGFP[Y66S], referred to as HAP1-dEGFP from hereon[53]. As transfection method we used "induced Transduction by Osmocytosis and Propanebetaine" (iTOP), a previously reported method for the intracellular delivery of recombinant proteins[54]. Next, the total RNA integrity was assessed using a Bioanalyzer. The total RNA profile of dEGFP expressing cells transfected with LbuCas13a and a dEGFP targeting guide RNA clearly revealed a degradation pattern mainly consisting of several novel fragments preceding the 18S ribosomal RNA (rRNA) peak (Fig. 1a). These fragments did not appear in the total RNA of wildtype HAP1 cells transfected with LbuCas13a and a dEGFP targeting guide RNA. We calculated the area under the curve of these LbuCas13a induced degradation fragments and found that their abundance peaked at 100 to 200 min after transfection, with some cleavage fragments remaining at 24 h after transfection (Fig. 1b). None of the other Cas13 orthologs were able to generate such a distinct change to the total RNA pattern for any of the target RNAs tested (Supplementary Fig. 2). Next we performed an LbuCas13a protein and guide RNA concentration gradient, ranging from 1 to 15 μM and using an equimolar ratio of protein and guide RNA (Supplementary Fig. 3). Collateral cleavage was evident in the total RNA profile from 1 μM and increased logarithmically with the concentration. Because 15 μM yielded the strongest collateral effect and further increases in concentration probably would only give a minimal improvement, we used 15 μM of protein and guide RNA in all further experiments.

To specifically assess the cleavage of the target transcript and a collateral transcript, we used a modified RT-qPCR strategy designed to quantify transcript integrity[55–57], here named 5′−3′ RT-qPCR. In the 5′−3′ RT-qPCR assay, cDNA is created by reverse transcription using oligo-dT primers, driving cDNA synthesis from the 3′ end of the mRNA transcript (Fig. 1c). Thus, transcript degradation results in a 3′ bias of the cDNA. Consequently, a 5′ located primer pair is expected to return higher Cq values than a primer pair located at the 3′ end of the transcript. To validate this assay, we first transfected cells with RNase A protein. This indeed resulted in the expected Cq value increase of the 5′ primer pairs relative to the 3′ primer

pairs (Supplementary Fig. 4a). Using the 5′−3′ RT-qPCR assay, we observed that LbuCas13a was able to degrade both target RNA (dEGFP) and collateral RNA (GAPDH) (Fig. 1d). While target RNA cleavage seemingly started almost immediately, the collateral GAPDH cleavage started later and peaked at around 100 min after transfection. Collateral RNA integrity returned to normal from about 500 min after transfection. dEGFP targeting with LwaCas13a, PspCas13b, and BzCas13b also resulted in a significant relative increase of the 5′ Cq values of the dEGFP transcript, indicating that these orthologs degraded the dEGFP target RNA in HAP1-dEGFP cells (Supplementary Fig. 4b). RspCas13d did not show significant target RNA cleavage. LwaCas13a, BzCas13b, and PspCas13b showed a minimal increase in non-target RNA 5′ Cq values compared to the non-targeting control (Supplementary Fig. 4b), suggesting low levels of collateral cleavage; however, these increases were not statistically significant. RspCas13d did not show an increase of collateral RNA 5′ Cq values (Supplementary Fig. 4b).

Taken together, we observed that under these specific conditions (i.e., iTOP RNP delivery, HAP1 cells, for these specific guide RNAs), LbuCas13a exhibited the highest collateral RNA cleavage activity both in vitro and in cells. Therefore, we used LbuCas13a for all further experiments. Collateral cleavage inside cells has also been reported for some of the other orthologs compared here[49–51]; however, we could not find strong evidence for collateral cleavage with these orthologs. This further highlights the situational nature of collateral cleavage by Cas13 in human cells[58], which we further discuss in Supplementary Note 1.

### Different RNP delivery methods support LbuCas13a collateral cleavage

We next examined whether collateral cleavage could also be induced using different delivery methods, and whether it depended on targeting exogenous RNA. First, we electroporated HAP1 cells with LbuCas13a and a guide RNA targeting one of the highly abundant endogenous transcripts RPS19, GAPDH, or 18S rRNA. Delivery by electroporation resulted in strong total RNA degradation for all three guide RNAs, demonstrating that collateral cleavage is not the result of the iTOP delivery method per se, and does not require targeting an overexpressed exogenous transcript (Supplementary Fig. 5a).

To explore if the collateral RNA cleavage also occurs when LbuCas13a is overexpressed in the cell, we transfected an LbuCas13a expression plasmid into HAP1-dEGFP cells. The transfected cells showed a clear LbuCas13a band on western blot (Supplementary Fig. 5b), despite an average transfection efficiency of only 9.2%. We enriched successfully transfected cells by FACS two days after plasmid transfection. Both the unsorted and enriched cells were transfected with a dEGFP-targeting guide RNA after allowing the enriched cells to recover overnight from sorting. No collateral cleavage pattern was observed in the protein expressing cells transfected with guide RNA (Supplementary Fig. 5c). Altogether, our data demonstrate that LbuCas13a is capable of collateral cleavage, targeting both exogenous and endogenous transcripts, and using two different RNP delivery methods. However, no collateral cleavage was observed when the LbuCas13a protein was expressed.

### Collateral cleavage occurs in all tested cell lines

Since previous reports have demonstrated that Cas13 activity can be cell-line dependent[58], we next explored whether collateral RNA cleavage by LbuCas13a RNP delivery was also cell-type dependent. We used iTOP to transfect the chronic myeloid leukemia derived line HAP1, the rhabdomyosarcoma line RD, the osteosarcoma line U2OS, and the retinal pigment epithelia line ARPE19 with LbuCas13a targeting the highly abundant endogenous transcripts RPS19, GAPDH, or 18S rRNA. All four cell lines demonstrated robust collateral cleavage after targeting any of these target transcripts (Fig. 2a). Across all four cell lines, 18S rRNA targeting resulted in the most robust collateral RNA cleavage activity, followed by GAPDH and RPS19, respectively. Notably, the degradation pattern was exactly the same in all conditions and does not depend on the target RNA or cell line. This consistent RNA degradation pattern allowed us to quantify the total RNA

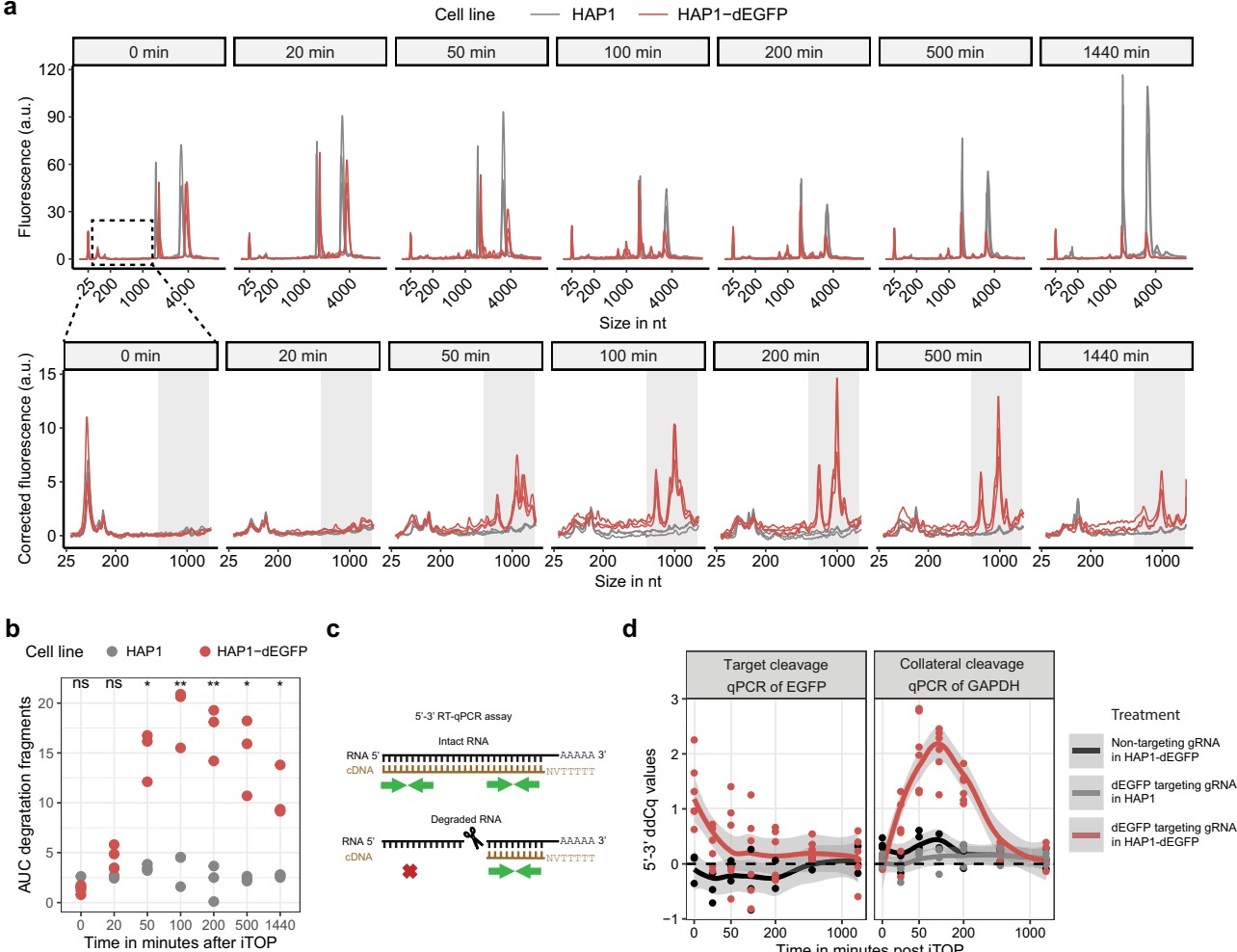

**Fig. 1 | LbuCas13a exhibits strong collateral cleavage activity in human cells.**
**a** Total RNA profiles at various timepoints after transfection of the LbuCas13a RNP
($n = 3$). Both HAP1 and HAP1-dEGFP cells were transfected with LbuCas13a and a
dEGFP targeting guide RNA. The bottom row is zoomed in to the region indicated
by the dotted box in the 0 min panel. Fluorescence values of the bottom row are
corrected for differences in amount of loaded RNA. **b** Area under the curve values of
the RNA fragments in the region outlined by the grey shaded areas in the bottom row
of (**a**) ($n = 3$). P-values were calculated by two-sided unpaired Welch's t-test (ns not
significant, * = p-value < 0.05, ** = p-value < 0.01). **c** Schematic depiction of the $5'$
$-3'$ RT-qPCR assay. Green arrows indicate the $5'$ and $3'$ primer pairs used for qPCR.

Red cross indicates no amplification is possible when there is a break between the
poly-A tail and the $5'$ primer pair. RNA is depicted in black, cDNA in brown. **d** $5'-3'$
RT-qPCR assay time course assessing degradation of target (*dEGFP*) and collateral
RNA (*GAPDH*) after transfection of HAP1 or HAP1-dEGFP with LbuCas13a and a
guide RNA ($n = 3$ for the non-targeting guide RNA in HAP1-dEGFP and dEGFP
targeting guide RNA in HAP1, $n = 6$ for dEGFP targeting guide RNA in HAP1-
dEGFP). The lines and their grey bands show ggplot2's geom_smooth default LOESS
regressions and their 95% confidence intervals[92]. Since wildtype HAP1 control cells
do not contain dEGFP, no $5'-3'$ RT-qPCR assay on dEGFP was performed for
these cells.

degradation by taking the area under the curve of the region where most of
the cleaved RNA fragments occur (See Fig. 2a). HAP1 cells consistently
demonstrated significantly more total RNA degradation than the other
three cell lines (Fig. 2b). Next, we utilized the $5'-3'$ RT-qPCR assay to
further quantify the collateral cleavage effect in these samples. RNA
degradation was presented by both an overall increase in Cq values com-
pared to untreated cells as well as an increased difference between the $5'$ and
$3'$ primer pair (Fig. 2c), with HAP1 again showing the biggest effect.
HAP1 cells are known as near-haploid cells; however, they diploidize
quickly in just several passages[59]. The HAP1-dEGFP cells we used in this
study are indeed a mixture of haploid and diploid cells (Supplementary
Fig. 6b). It could be haploid cells are more sensitive to collateral activity due
for example, not being able to replace lost RNA as quickly as diploid cells.
Another possible reason for the higher collateral activity in HAP1 cells could
be a higher RNP delivery efficiency, as iTOP was developed using this cell
type[54].

Several reports have shown specific RNA knockdown with Cas13 used
HEK293T cells[11,14,30], while others explicitly reported low or even the absence

of collateral cleavage in this cell type[45,49], suggesting that Cas13 may be less
active in HEK293T cells. Using electroporation to deliver LbuCas13a pro-
tein and guide RNA into HEK293T cells, we observed the characteristic
RNA degradation pattern upon targeting *GAPDH* and *18S rRNA* (Sup-
plemental Fig. 6c), indicating that HEK293T cells were not completely void
of LbuCas13a collateral cleavage activity. However, in agreement with above
mentioned publications, the extent of the collateral cleavage was decreased
compared to HAP1 when targeting *GAPDH* and even undetectable when
targeting the highly abundant *RPS19* transcript.

In conclusion, LbuCas13a exhibits collateral RNA cleavage in all tested
cell lines. However, the activity varies between cell lines, which could be
caused by differences in RNP delivery efficiency, cell ploidy, or other bio-
logical factors such as Cas13 protein and guide RNA degradation rates.

**RNA targeting with LbuCas13a induces cell death via apoptosis**
Over the course of our experiments, we noticed that a day after targeting
dEGFP with LbuCas13a cells were rounding up and detaching from the
bottom of the plate (Fig. 3a), implying loss of viability. To examine this

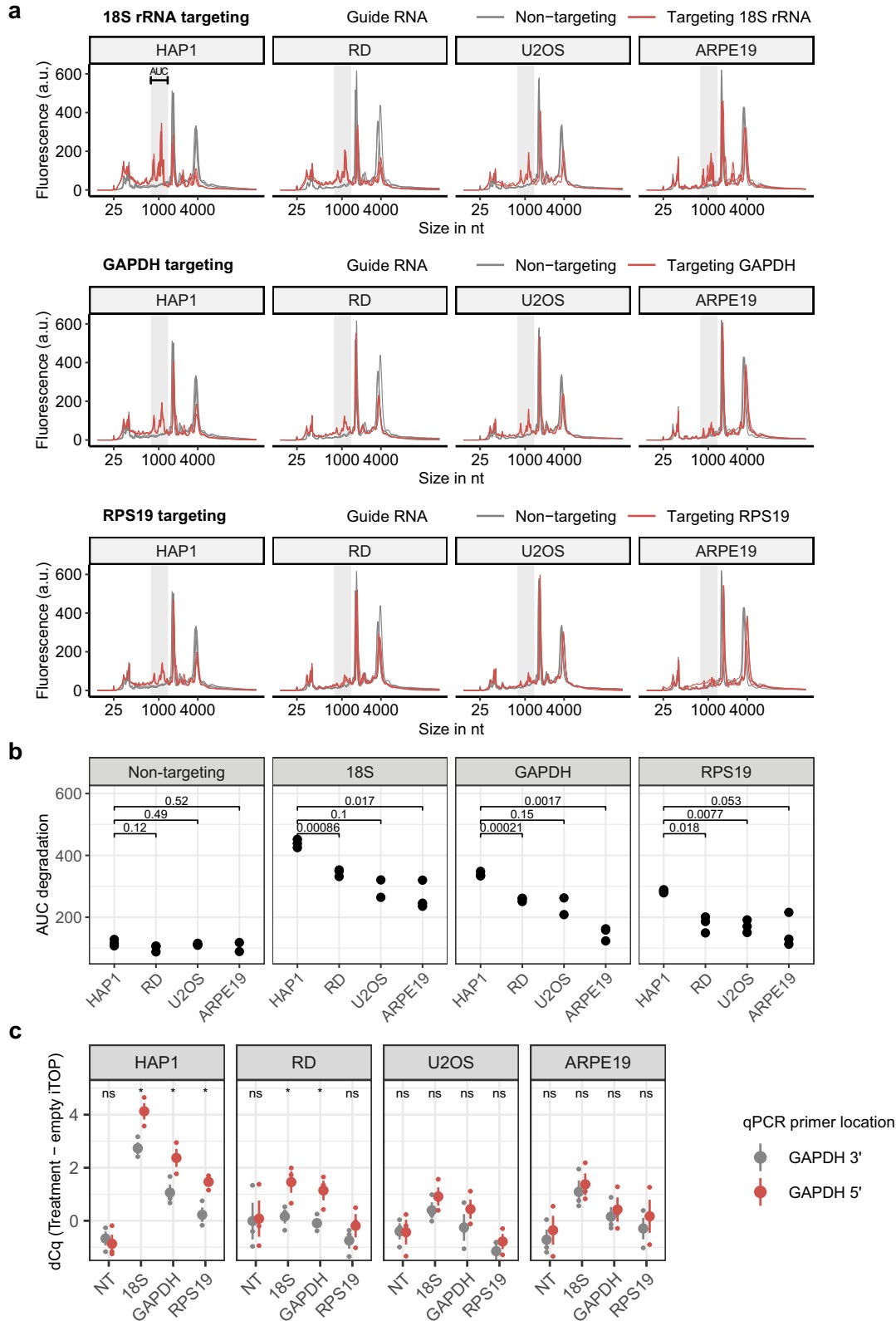

**Fig. 2 | LbuCas13a displays collateral cleavage in different cell lines. a** Total RNA profiles after targeting the endogenous transcripts 18S rRNA, GAPDH, and RPS19, respectively. Total RNA was isolated 100 min after RNP transfection. Fluorescence values from the Bioanalyzer were normalized to the total area under the curve of each sample. $n = 3$, except for the U2OS 18S rRNA, U2OS GAPDH and ARPE19 non-targeting samples, where $n = 2$. **b** Quantification of total RNA cleavage in different cell lines, by taking the area under the curve of the region outlined by the grey shaded areas in (**a**). $n = 3$, except for the U2OS 18S rRNA, U2OS GAPDH, and ARPE19

non-targeting samples, where $n = 2$. *P*-values were calculated by two-sided unpaired Welch's *t*-test. **c** RT-qPCR assays on samples from (**a**) ($n = 3$). Here, dCq values (Cq treatment - Cq empty iTOP transduction) are shown for both a 3′ and 5′ located primer pair. The big dot and line show mean and standard error of the mean, small dots are individual samples. *P*-values were calculated by comparing GAPDH 5′ with GAPDH 3′ using an two-sided unpaired Welch's *t*-test (ns not significant, * = *p*-value < 0.05).

**Fig. 3 | Cells enter apoptosis in response to collateral RNA cleavage. a** Loss of HAP1-dEGFP cell viability 24 h after treatment with LbuCas13a and a dEGFP targeting guide RNA. Scale bar is 50 μm. **b** Target RNA-specific induction of apoptosis in HAP1 cells by LbuCas13a. Total number of Annexin V (early apoptosis marker) and CellTox Green (dead cell marker) positive cells during live imaging after transfection. CTRL is an empty transfection. SP1, SP2, and SP3 are transfections with LbuCas13a and three different guide RNAs all targeting dEGFP. Images were taken every 15 min. The number of dye positive cells was quantified using CellProfiler[90]. The number of Annexin V positive cells varied significantly between cell lines (Repeated-measures ANOVA, $F_{1,15} = 9.565$, $p = 0.00743$) and treatments (Repeated-measures ANOVA, $F_{3,15} = 4.812$, $p = 0.01530$). The number of CellTox Green positive cells was not significantly different between cell lines (Repeated-measures ANOVA, $F_{1,15} = 1.093$, $p = 0.3123$) and treatments (Repeated-measures ANOVA, $F_{3,15} = 1.225$, $p = 0.3350$). $n = 3$, expect for the CTRL, SP1, and SP2 treatments in HAP1, where $n = 2$.

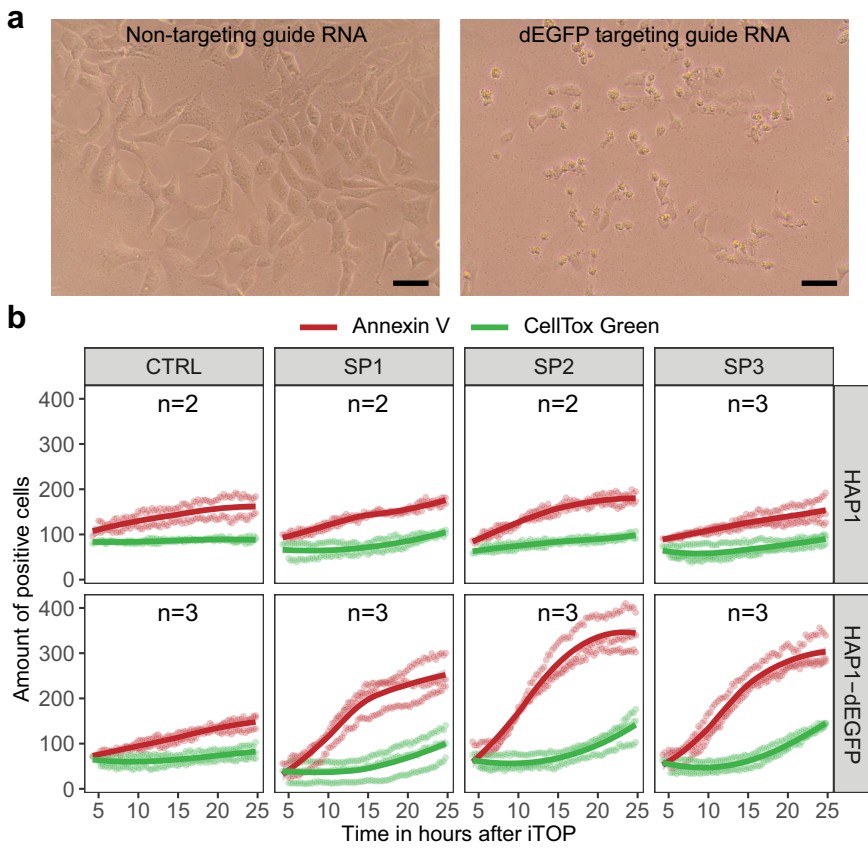

further, we performed live cell imaging with Annexin V as a marker for early apoptosis and CellTox Green to stain for dead cells[60]. When cells die by necrosis, they become positive for both these dyes simultaneously. However, if they go into apoptosis Annexin V staining precedes CellTox green staining. We transfected LbuCas13a and one of three different guide RNAs targeting *dEGFP* into either wildtype or *dEGFP* expressing cells HAP1. In wildtype cells the number of Annexin V and CellTox green positive cells was similar to the empty transfection control for all three *dEGFP* targeting guide RNAs (Fig. 3b). However, in HAP1-dEGFP the number of Annexin V positive cells rapidly increased from approximately 7.5 h after transfection, followed by an increase in CellTox Green positive cells approximately 15 h after transfection (Fig. 3b and Supplementary Movies 1 and 2). This shows that LbuCas13a is capable of inducing cell death, and that this is specific to cell expressing the target RNA. Furthermore, it demonstrates that the targeted cells go into apoptosis, not necrosis.

**Application of LbuCas13a as a cell selection tool**

Given that target-activated collateral RNA cleavage by LbuCas13a can induce cell death, we wondered whether we could exploit this property to eliminate cells that express a specific target RNA from a heterogeneous cell population (Fig. 4a). To test this, we mixed fluorescent EGFP-expressing HAP1-EGFP cells with wildtype HAP1 cells and transfected this mixed population with LbuCas13a and a guide RNA targeting *EGFP*. It is important to note that this *EGFP* targeting guide is the same guide RNA used to target dEGFP above, and does not cause collateral RNA cleavage or reduced viability in wildtype cells (Figs. 1b, d and 3b). We then measured the percentage of EGFP positive cells by FACS over time. The percentage of EGFP positive cells rapidly decreased when targeting *EGFP* (Fig. 4b), with 36.3% EGFP positive cells remaining at day 6 compared to more than 60% EGFP positive cells when using a non-targeting guide. The decrease in EGFP positive cells was maintained over the entire six-day assay, showing that it was not due to temporary knockdown of the *EGFP* transcript itself, as the *EGFP* transcript integrity was largely restored after 500 min to 24 h after RNP transfection (see Fig. 1d). Next, we examined

whether multiple sequential selection events could further eliminate the EGFP positive cell population (Fig. 4c). We performed three serial transfections at 2-day intervals. Every additional selection round decreased the percentage of EGFP positive cells in the population by approximately 20 percentage points. From the above data, we conclude that Cas13-based cell-selection is able to reduce the target cell population and is repeatable. Target RNA expression level theoretically limits the number of active LbuCas13a molecules and therefore the amount of collateral cleavage, which could influence the efficiency of cell selection. To test this, we FACS sorted a heterogeneous population of *EGFP* expressing cells into cell populations with different levels of EGFP intensity (Fig. 4d and Supplementary Fig. 7a). RT-qPCR confirmed *EGFP* expression levels increased with EGFP intensity (Supplementary Fig. 7b). We then tested the depletion efficiency of these cells by mixing them with wildtype HAP1 cells and targeting *EGFP* with LbuCas13a. In parallel, we also performed RT-qPCR on these cell lines to quantify their *EGFP* expression at the time of the experiment. We found a strong correlation between the expression level of the target RNA and the depletion of EGFP positive cells (Fig. 4e), suggesting that activating more LbuCas13a proteins results in more collateral RNA cleavage and a higher probability of inducing cell death. However, the depletion efficiency seems to plateau at the highest levels of target RNA expression. At this point, the RNP delivery efficiency may have become the limiting factor. We next examined whether the same strategy could be used to positively select for cells that underwent successful gene editing. For these experiments, we used a HAP1-EGFP^Afluor cell line containing a 9 base pair deletion in *EGFP*, rendering it non-fluorescent (Supplementary Fig. 7c)[53]. Using this system, we previously reported Cas12a-mediated correction of EGFP fluorescence, with an efficiency of about 5%[53]. When we employed LbuCas13a and a guide RNA that targets the unedited cells as a cell selection tool, the percentage of EGFP expressing cells increased to around 15% (Supplementary Fig. 7d). Thus, a single round of selection using LbuCas13a resulted in a threefold enrichment.

Finally, we investigated whether cell selection with LbuCas13a can be used to eliminate cancer cells by targeting an endogenous oncogenic

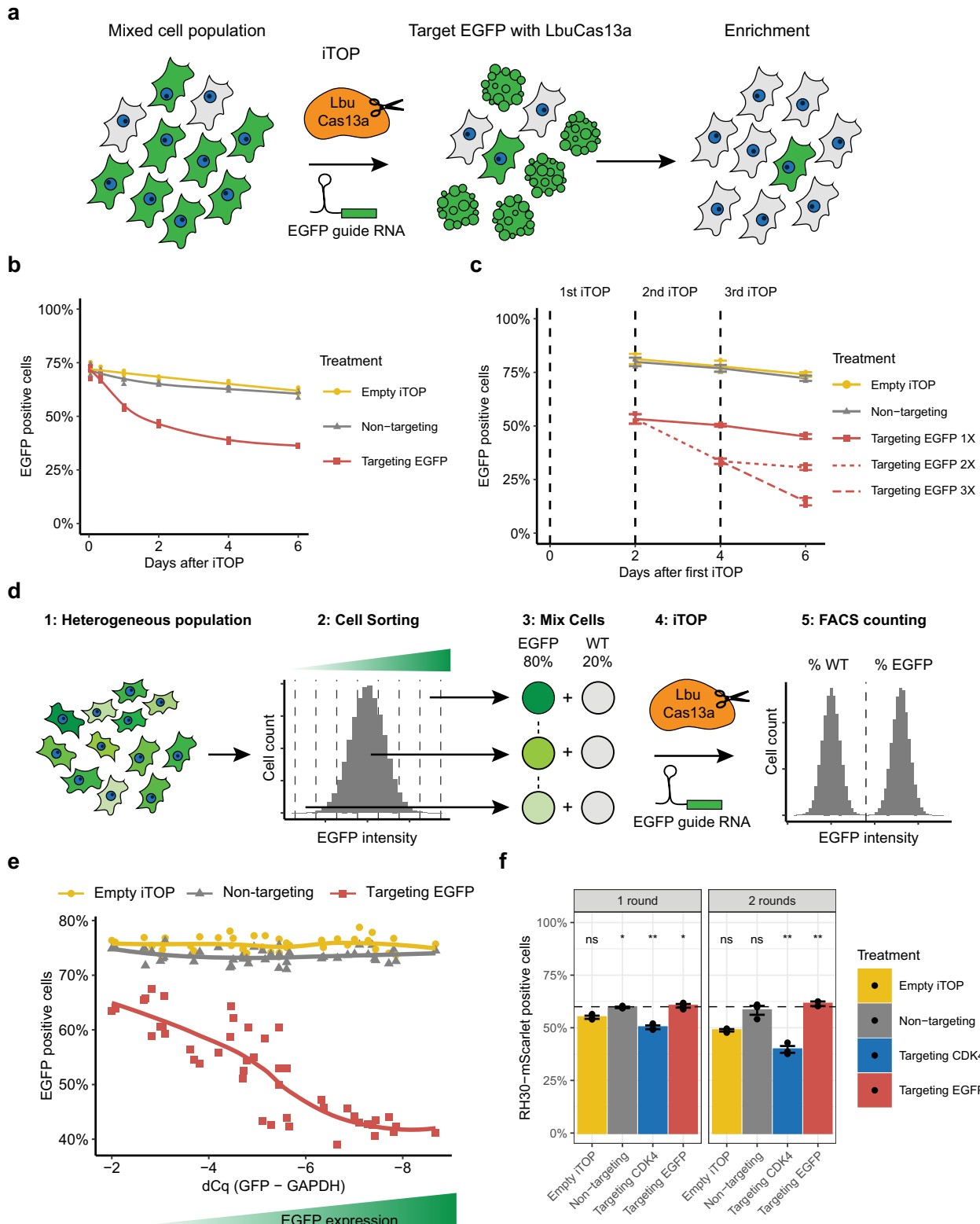

transcript. We used the rhabdomyosarcoma line RH30 as a model. This line contains a genomic amplification of the 12q13-14 region, resulting in *CDK4* overexpression[61]. As a control we used the rhabdomyosarcoma line RD, which does not have this genomic amplification[61]. First we confirmed *CDK4* overexpression in RH30 by RT-qPCR (Supplementary Fig. 8a). RH30 had about 14.5 times higher *CDK4* expression than RD. Since we have shown

that high target RNA expression is required for cell selection with Lbu-Cas13a (Fig. 4e), we wondered if we could find a *CDK4* targeting guide RNA that generated substantial collateral RNA cleavage in RH30, but not in RD. Therefore, we compared the collateral activity of five different (SP1-SP5) *CDK4* targeting guide RNAs in both cell lines (Supplementary Fig. 8b, c). Three guides were active in RH30, with SP1 showing the highest collateral

**Fig. 4 | LbuCas13a can be used as a cell selection tool. a** Schematic of the Cas13-based cell selection. **b** Depletion of EGFP positive HAP1 cells after targeting the *EGFP* transcript, as measured by FACS (*n* = 3). Trendlines show a generalized additive model using formula y ~ s(x, *k* = 6). Cells were mixed the day before iTOP aiming for ~75% EGFP positive cells. **c** Multiple rounds of targeting improved depletion of EGFP positive HAP1 cells, as measured by FACS (*n* = 3). Error bars show standard deviation. Cells were mixed the day before iTOP aiming for ~75% EGFP positive cells. **d** Schematic showing creation of cell lines with different EGFP expression levels and subsequent selection experiments. **e** Target RNA expression level determines cell selection efficiency (*n* = 3). Target RNA expression level was determined using RT-qPCR, and the percentage of EGFP positive cells remaining

after treatment was determined using FACS. Trendlines created by ggplot2's default LOESS regression. Cells were mixed the day before iTOP aiming for ~75% EGFP positive cells. **f** RH30-mScarlet or RD-EGFP cells can be selected against in a mixed RH30:RD population by targeting either CDK4 or EGFP transcripts, respectively (*n* = 3). The percentage of RH30-mScarlet cells at the start of the experiment is shown by the horizontal dashed line. The cell mixes were exposed to two rounds of treatment, with four days between the two treatment rounds. The percentage of RH30-mScarlet and RD-EGFP cells was assessed by FACS three days after each round. Error bars show the standard error. *P*-values were calculated by comparing each condition against the base-mean using a two-sided unpaired Welch's *t*-test (ns not significant, * <0.05, ** <0.01).

activity. All tested guide RNAs had no or minimal activity in RD. Therefore we selected the SP1 guide RNA for all further experiments. We then performed a titration of the RNP concentration, ranging from 7.5 to 60 μM (Supplementary Fig. 8de). The collateral activity in RH30 strongly increased with the RNP concentration, while in RD a small increase in collateral cleavage fragments was observed. Next, we performed a CellTiter-Glo assay to assess to effect of *CDK4* targeting on the viability of RH30 and RD, testing both the standard 15 μM and the increased 60 μM RNP concentrations (Supplementary Fig. 8f). At 15 μM, *CDK4* targeting resulted in a significantly lower viability in RH30 compared to RD. However, at 60 μM both lines showed a pronounced decline in viability and the difference in viability between RH30 and RD was no longer significant. Finally, we assessed whether *CDK4* targeting can be used to deplete RH30 cells from a mixed population. We combined RH30-mScarlet and RD-EGFP cells at an approximately 2:1 ratio and subjected them to two sequential rounds of *CDK4* or *EGFP* targeting (Fig. 4f). We observed a small decrease in the percentage of RH30 cells in the empty iTOP condition, suggesting that RD has a higher relative growth rate under these conditions. Surprisingly, treatment with a non-targeting guide RNA seemed to give a relative disadvantage to RD compared to the empty iTOP transfections. This could be due to an unintended non-targeting guide specific effect in this cell line, although we did not observe a negative effect of the non-targeting treatment on the viability of RD using the CellTiter-Glo assay (Supplementary Fig. 8f). As anticipated, selecting against RH30 by targeting *CDK4* resulted in the biggest decrease in RH30 cells, while selecting against RD by targeting *EGFP* resulted in the highest percentage of RH30 cells. Altogether, these proof-of-principle studies show the potential of Cas13 as a potent cell selection tool, including the enrichment of successfully edited cells and the depletion of cancer cells.

### Genes involved in the innate immune system are upregulated after collateral RNA cleavage

To gain further insight into the cellular response to the collateral RNA cleavage and the mechanism by which it triggers apoptosis, we performed RNA sequencing at 16 h after transfection. Based on the $5'−3'$ RT-qPCR data (Fig. 1d), at this timepoint most mRNA cleavage has been resolved, allowing us to focus on the cellular response following the collateral cleavage. STAR[62] aligned sequences at similar rates across all conditions, however, a higher percentage of the reads from the *dEGFP* targeting samples mapped to introns compared to the controls, consistent with a relative increase in immature nuclear transcripts generated as a result of cytoplasmic RNA depletion in the cell (Supplementary Fig. 9a). First, we performed principal component analysis (PCA) (Supplementary Fig. 9b). The PCA plot shows a clear separation between the *dEGFP* targeting samples and the controls, suggesting that collateral cleavage by LbuCas13a leads to substantial changes in the transcriptome. Indeed, in response to LbuCas13a activation, we found 806 significantly upregulated and 41 significantly downregulated genes (adjusted *P*-value < 0.05, |log$_2$(fold change)| > 1), while the exposure to exogenous protein and/or RNA without LbuCas13a activation led to a only small number of differentially expressed genes (Fig. 5a–c). The relatively low number of downregulated genes and high number of upregulated genes is perhaps somewhat counterintuitive in the context of collateral cleavage, but consistent with another report by Li and colleagues documenting the

cellular response to Cas13 collateral cleavage[50]. KEGG pathway enrichment analysis on the downregulated genes reveals enrichment of term related to fatty acid metabolism (Supplementary Fig. 9c). In agreement with Li et al., we observed that many of the upregulated genes we find are immediate early response genes, such as *JUN*, *FOS*, *EGR1*, *EGR2*, *EGR3*, *EGR4* and *ARC* (Fig. 5c). GO enrichment analysis of the significantly upregulated genes revealed an enrichment of many biological processes related to the innate cellular immune response (Supplementary Fig. 9d), and molecular functions related to chemokine and cytokine activity (Supplementary Fig. 9e). KEGG pathway analysis also showed enrichment for immune related pathways, such as viral protein interaction with cytokine and cytokine receptor, IL17 signaling pathway, TNF signaling pathway and NF-kappa B signaling pathway (Fig. 5d). These are known factors of the cellular immune response that are triggered, for example, by intracellular recognition of viral RNA and suggests that collateral RNA cleavage by LbuCas13a and subsequent increase in 5′ uncapped RNA and 3′ ends without poly-A tail triggers a similar immune response which ultimately leads to apoptosis.

### Collateral RNA cleavage leads to near-global mRNA loss

To get a transcriptome-wide overview of the transcripts affected by collateral RNA cleavage, we performed total RNA sequencing on samples collected at 50, 200, and 1440 min after iTOP transfection. Sequencing libraries were normalized to ERCC spike-in control RNAs to correct for potential widespread reductions in mRNA levels due to collateral RNA cleavage. This revealed a near-universal depletion of protein coding transcripts at 50 and 200 min after treatment with LbuCas13a and a targeting guide RNA (Fig. 6a), which was not apparent without ERCC spike-in RNA normalization (Supplementary Fig. 10a). The median log$_2$(fold changes) after *dEGFP* targeting with LbuCas13a for all protein coding transcripts were −0.66 and −1.18 at the 50 and 200 min timepoints respectively, while the fold changes in the control conditions remained centered at 0. This suggests an over 50% median loss of protein coding transcripts at 200 min after LbuCas13a activation. Interestingly, mitochondrially encoded mRNAs and nuclear non-coding RNAs such as *MALAT1*, snRNAs, and snoRNAs appeared unaffected (Fig. 6a and Supplementary Fig. 10b–d). Furthermore, transcript abundance and depletion due to collateral RNA cleavage seem to be correlated, with highly abundant transcripts being more sensitive. This trend has also been observed for the endogenous RNase RNASEL[63]. To find transcripts that are either less sensitive to collateral cleavage, or upregulated in response to collateral cleavage, we modelled this relationship between transcript abundance and depletion by performing a linear regression (Fig. 6a). Next we selected transcripts >1 log$_2$(fold change) above this regression line at 50 min, and >2 log$_2$(fold change) above the regression line at 200 and 1440 min after iTOP. This identified 68, 255, and 224 upregulated and/or Cas13 insensitive mRNAs at 50, 200, and 1440 min after the *dEGFP* targeting treatment, respectively. Immediate early stress response genes such as *FOS*, *JUN*, *FOSB*, *EGR1*, and *EGR2* were present at all three timepoints, suggesting cells have an immediate and sustained stress response to the collateral RNA cleavage. Besides these immediate early genes, genes related to the response to incorrectly folded proteins were enriched at 50 min after iTOP (Supplementary Fig. 10e). GO biological processes related to nucleosome assembly were enriched at 200 min after iTOP, due to 14 histone genes being present in the set (Supplementary Fig. 10f). The genes

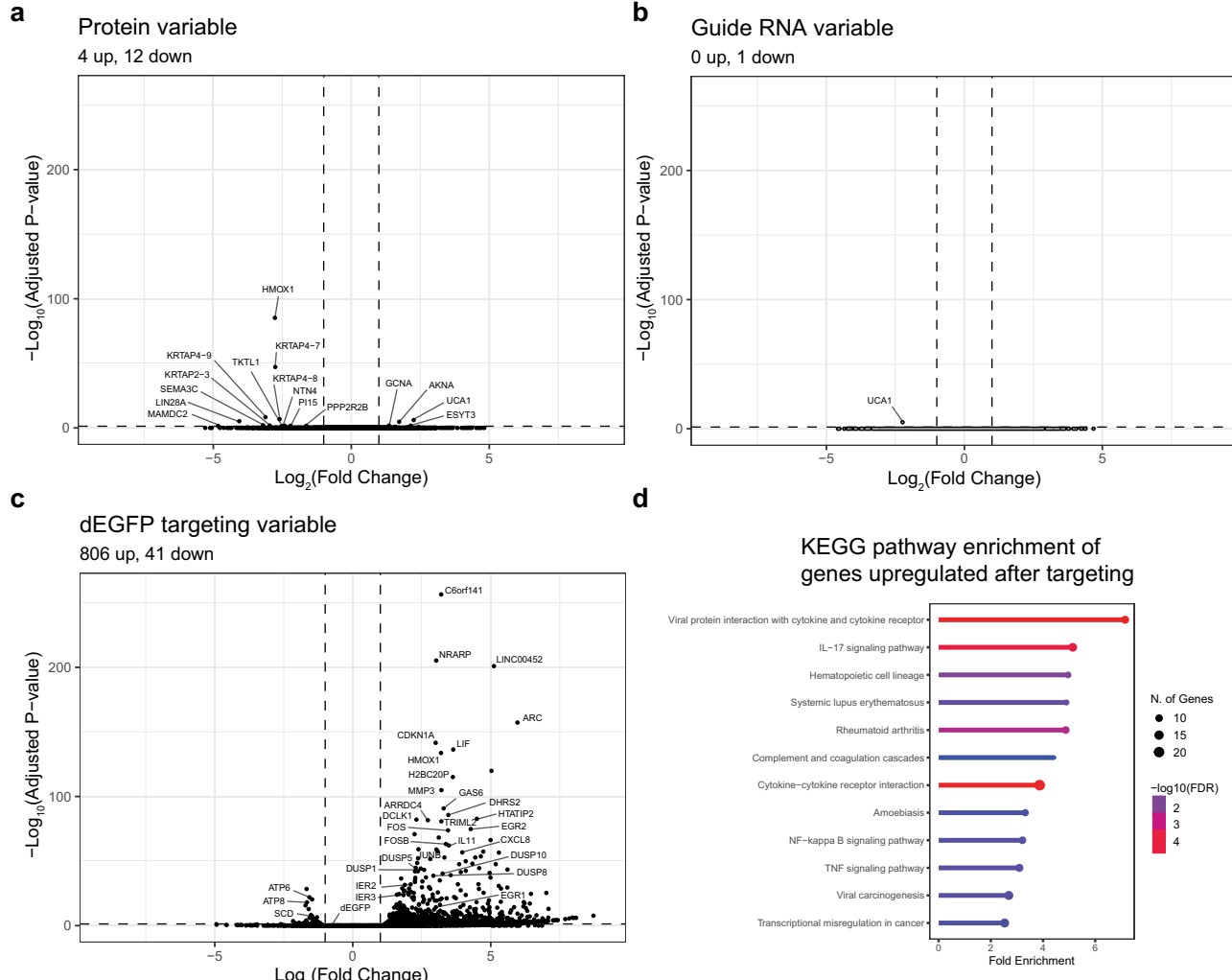

**Fig. 5 | Innate immune response genes are upregulated in response to collateral RNA cleavage. a–c** Differential gene expression in response to transfections with LbuCas13a protein (**a**), guide RNA (**b**), or LbuCas13 protein and a targeting guide RNA (**c**). Fold changes and *P*-values were calculated using DESeq2, see "Methods"

($n = 3$). Dashed lines indicate adjusted *P*-value = 0.05 and |Log$_2$(Fold Change)| = 1. **d** A ranked list of enriched KEGG pathways in the significantly upregulated genes in response to dEGFP targeting with LbuCas13a (adjusted *P*-value < 0.05 & log$_2$(fold change) > 1). Plot was generated using ShinyGo[88].

identified at 1440 min after iTOP most resembled the upregulated genes in the RNA-seq experiment performed at 16 h after treatment described above. Genes related to the IL17 pathway, MAPK pathway, and TNF pathway were enriched (Supplementary Fig. 10g, h).

**Characterizing collateral RNA cleavage using nanopore sequencing**

We decided to use Nanopore sequencing to further explore the identity of the RNAs that are subject to collateral cleavage, where RNAs get cleaved by LbuCas13a, and the dynamics of the collateral cleavage process. HAP1-dEGFP cells were transfected with LbuCas13a protein and either a non-targeting or a *dEGFP* targeting guide RNA. Nanopore read lengths of *dEGFP* targeting samples were much shorter than control samples (Supplementary Fig. 11c), containing markedly fewer reads longer than 1000 nt, suggesting widespread collateral RNA cleavage. Consistent with the total RNA sequencing above, we found that the number of protein coding transcripts decreased relative to all other transcripts biotypes when targeting *dEGFP*, although this difference was not significant (Supplementary Fig. 12b). In agreement with the total RNA-sequencing experiment above, the abundance and read length of reads mapped to protein coding transcripts greatly decreased at 50 and 200 min after transfection (Fig. 6b and Supplementary Fig. 12c), whereas most reads in the empty

transfection and non-targeting controls were full length transcripts. This suggests that the vast majority of mRNAs are subjected to collateral cleavage by LbuCas13a. Next, we explored the dynamics of cleavage of individual transcripts by calculating their relative transcript coverage. We found that at 50, 200, and 500 min after transfection, the overall coverage of *dEGFP-IRES-PuroR* is much lower for the *dEGFP* targeting samples than the control conditions (Fig. 6c). Additionally, the target RNA shows a decrease in coverage at the exact location of the protospacer at all time-points (Fig. 6c, dotted lines), which fits with the 5′−3′ RT-qPCR data that showed target RNA cleavage immediately after transfection. This suggests LbuCas13a cleaves in, or very near the protospacer, which is surprising since others have shown that Cas13 cleavage sites do not depend on protospacer position[9–11]. Alternatively, the protospacer specific dip in coverage may be a technical artifact, for example, due to LbuCas13a still being bound to the target RNA after RNA purification, preventing reverse transcription at this location. Besides the coverage dip around the pro-tospacer, additional dips in coverage were observed, which were con-sistent between timepoints and replicates. We suspected that these dips are LbuCas13a cleavage sites. Non-target mRNAs and cytoplasmic non-coding RNAs also show these dips in coverage at specific positions, indicating collateral cleavage of these transcripts (Fig. 6d and Supple-mentary Figs. 13 and 14). We did not observe any differences in coverage

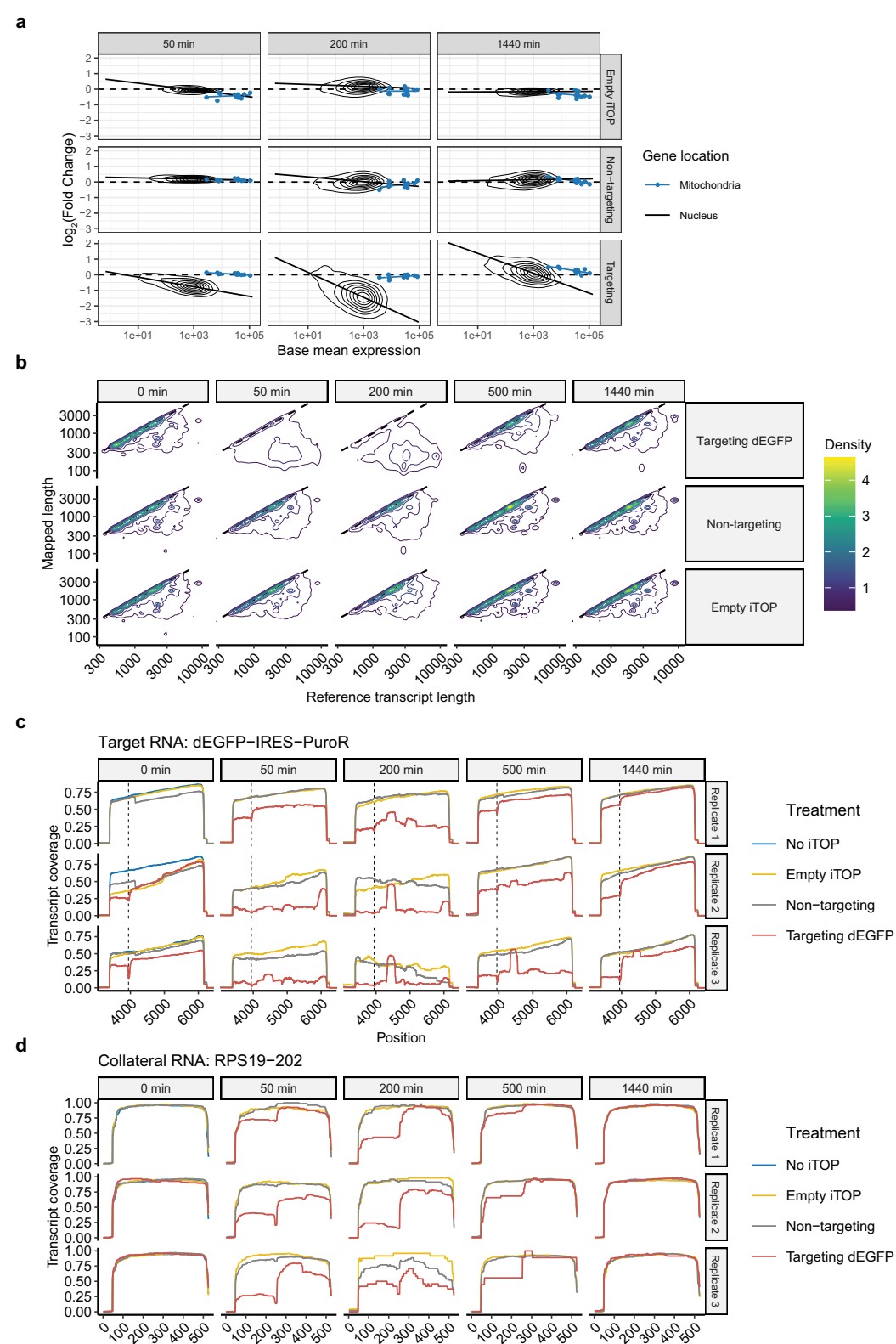

between targeting and control samples for small nuclear or mitochondrial RNAs (Supplementary Fig. 15). The dips in transcript coverage tend to be 10–20 nucleotides wide, so we cannot discern the exact cleavage position (Fig. 7a). In vitro, LbuCas13a was shown to prefer cleaving at uracil residues in or near single-stranded loop structures[21]. This fits with our data, as transcript coverage dips often map to predicted stem-loop

structures, with uracil containing single-stranded loops on the 5′ side of the coverage dip (Fig. 7b), suggesting that the cleavage preferences of LbuCas13a in cells is similar to its preferences in vitro. These results indicate that LbuCas13a cleavage preferences follow the same pattern in human cells as observed in vitro. Next, we manually annotated 100 potential cleavage sites, assuming that the Uracil closest to the 5′ side of the

**Fig. 6 | Total RNA and Nanopore sequencing reveals near-global mRNA loss after collateral RNA cleavage by LbuCas13a. a** Total RNA sequencing shows near-universal mRNA loss after collateral RNA cleavage. Mitochondrially encoded mRNAs (in blue) appear unaffected. Density plots of the log₂(fold change) versus the base mean expression for all protein-coding transcripts in response to an empty iTOP transfection, transfection with LbuCas13a and a non-targeting guide RNA, or with LbuCas13a and a dEGFP targeting guide RNA. Libraries were normalized to ERCC spike-in control RNAs before differential gene expression with DESeq2 (see "Methods"). Columns show minutes after iTOP transfection. Experiments performed with HAP1-dEGFP. $n = 3$ except for the empty iTOP condition at 50 min after iTOP where $n = 2$ due to one sample failing library prep and sequencing QC. **b** Nanopore sequencing shows near-universal mRNA loss after collateral RNA cleavage. Density plots of mapped length versus reference transcript length of all reads were mapped to protein coding transcripts, excluding histone genes and *WDR74*. Columns show minutes after transfection. Experiments were performed with HAP1-dEGFP. Density plots are based on three biological replicates, except

0 min Targeting dEGFP where $n = 2$. The density around y = x for the targeting condition at 50 and 200 min originates almost entirely from replicate 1, where we saw the lowest activity (Supplementary Fig. 11a). See Supplementary Fig. 12c for separate densities per replicate. **c** Nanopore sequencing transcript coverage of the target transcript dEGFP. Reads were aligned to the entire plasmid sequence used to make this cell line (Supplementary Table 4, pEF-1a_dEGFP-Y66S-IRES-PuroR-WPRE). The *dEGFP* start codon and *PuroR* stop codon are at position 3415 and 5316, respectively. The dashed vertical lines indicate the protospacer position. Columns show minutes after transfection. The *dEGFP* targeting sample at 0 min after transfection of replicate 1 was excluded due to failed library preparation. For all other conditions, $n = 3$. **d** Nanopore sequencing transcript coverage of *RPS19-202*, a collateral RNA. *RPS19-202* appears to have one major cleavage site. Columns show minutes after transfection. The *dEGFP* targeting sample at 0 min after transfection of replicate 1 was excluded due to failed library preparation. For all other conditions, $n = 3$.

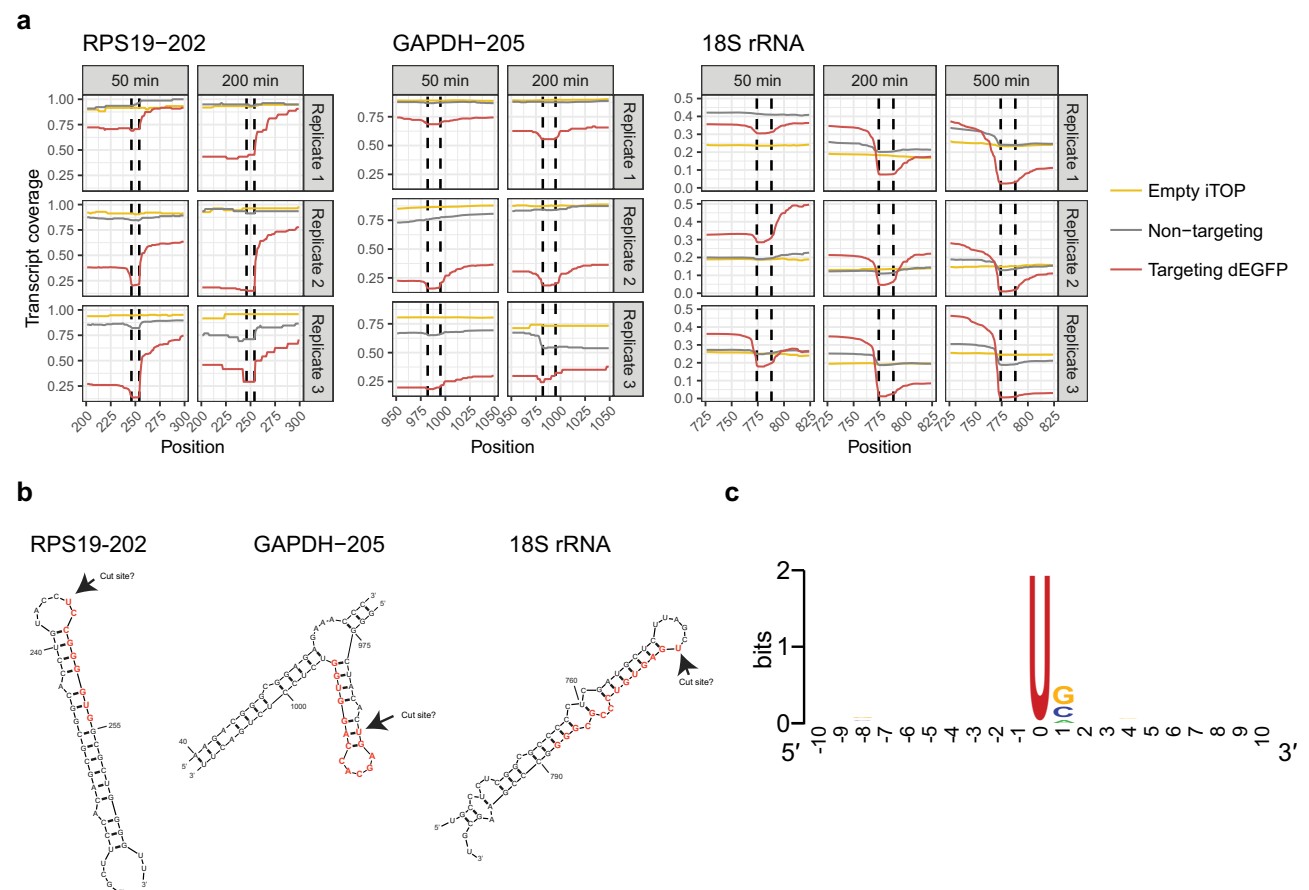

**Fig. 7 | LbuCas13a collateral RNA cleavage sites in human cells. a** Nanopore sequencing transcript coverage of non-target RNAs, zoomed in at putative cleavage sites ($n = 3$). Columns show minutes after transfection. Vertical dashed lines indicate putative cleavage sites. **b** The predicted RNA structures at the putative cleavage sites from (**a**). Red letters are the approximate locations of the coverage dips, corresponding to putative cleavage sites indicated by the dashed lines in (**a**). RNA structures for *RPS19-202* and *GAPDH-205* are the top predictions of Mfold[93] using the default settings. *18S rRNA* structure provided by RNAcentral.org[94] (URS0000726FAB_9606). **c** Sequence logo around 100 manually annotated cleavage sites. The U at position 0 was enforced during potential cleavage site selection. Logo made using WebLogo[95].

coverage dip was the cleavage position. This revealed a US (S = G or C) dinucleotide preference (Fig. 7c). Taken together, the total RNA sequencing with spike-in control RNAs and the Nanopore sequencing provide further evidence of collateral RNA cleavage in human cells by LbuCas13a and suggest that both mRNAs and cytoplasmic ncRNAs are near universally subjected to collateral cleavage. Additionally, it suggests that LbuCas13a cleaves RNA at specific positions. We propose that the in vitro preference of LbuCas13a for cleaving at uracil residues in single-stranded RNA loops is preserved in human cells. However, further experiments are needed to determine exact cleavage positions and structural preferences.

## Discussion

In this work, we revealed the temporal dynamics of both target and collateral RNA cleavage by LbuCas13a, as well as the cellular responses to this collateral RNA cleavage (Fig. 8). The collateral RNA cleavage activity of Cas13 is well established in vitro[9–13,15,21] and in bacteria[9,10,12,22]; however, in eukaryotes the situation is less clear. While many groups report that no collateral cleavage seems to occur[11,14,30,64,65], others have recently run into unexpected cytotoxicity and/or apparent collateral RNA cleavage when using CRISPR-Cas13 in eukaryotic cells[45,47–52]. From these reports and our work, several factors have emerged that are likely important in determining if and how

**Fig. 8 | A timeline of events after RNA targeting with LbuCas13a in cells. a** This plot provides a summary of the timeline of events after RNA targeting with LbuCas13a. The target RNA cleavage timing takes into account both Figs. 1d and 6c. While Fig. 1d suggests target cleavage peaks at 0 min after delivery, Fig. 6c suggests it peaks at 50–200 min after delivery. Collateral RNA cleavage timing is based on Figs. 1d and 6a, b and d. For ribosomal RNA degradation, see Fig. 1a and Supplementary Fig. 11a. The early apoptosis timeline is based on Fig. 3b. The cell death timeline is based on Figs. 3b and 4b.

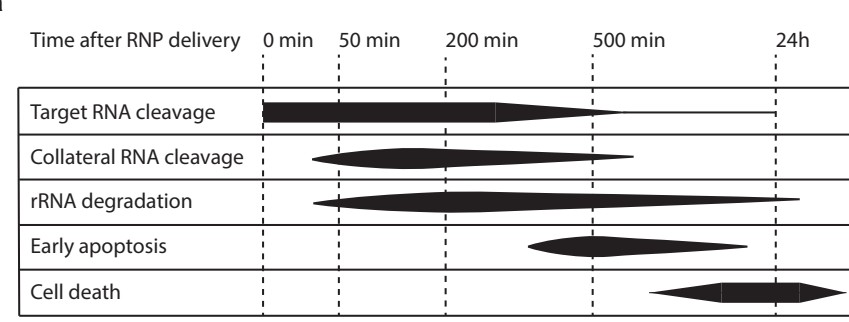

much collateral cleavage occurs, namely the collateral activity of the Cas13 ortholog, the intracellular RNP concentration, the target RNA expression level and cell line specific differences[58]. When comparing the collateral activity of several Cas13 orthologs in vitro, in our hands Lbu-Cas13a was the most active ortholog. However, different orthologs may have different preferences for factors such as the assay buffer, the protospacer sequence[64,66] and the cleavage substrate[21]. When comparing the collateral RNA cleavage activities of these Cas13 orthologs in cells, only LbuCas13a exhibited robust collateral cleavage across a range of different guide RNAs. Recent reports have documented collateral cleavage or toxicity when using LwaCas13a[45,47], or RfxCas13d[49–51]. Unfortunately, we were unable to produce active RfxCas13d protein. We did not find strong evidence of collateral RNA cleavage activity in human cells by LwaCas13a, which may be due to differences in delivery method, cell line, target RNA, or spacer sequence (Supplementary Note 1).

We found a clear correlation between the RNP concentration applied during delivery and the resulting collateral RNA cleavage. This suggests that the intracellular RNP concentration is an important factor, and that an efficient delivery method is important to generate substantial collateral RNA cleavage. Recent reports showed that besides the canonical target RNA activated RNase activity, Cas13 also exhibits both guide RNA independent and guide RNA dependent but target RNA independent RNase activity[67,68]. This was shown to have a negative effect on cell viability, and thus may provide selection pressure against persistent Cas13 expression. Interestingly, recent reports of collateral cleavage by RfxCas13d in eukaryotic cell lines used transient plasmid transfection[50,51], while publications finding no collateral cleavage by RfxCas13d used stable lentiviral integration[64,65]. We hypothesize that these non-canonical RNase activities of Cas13 may limit the intracellular Cas13 contraction when Cas13 is expressed, which could also explain why we were unable to detect collateral cleavage when expressing LbuCas13a.

Here we showed that high target RNA expression was required for the induction of detectable levels of collateral RNA cleavage and for negative cell selection against target RNA expressing cells. All target RNAs used in this study are in the top 1% of expression levels. We demonstrated that the requirement for high target RNA expression can be utilized to selectively induce collateral RNA cleavage in a cancer cell line overexpressing *CDK4*. This strategy may be more broadly applicable, as genomic amplifications and overexpression of amplified (onco)genes are relatively common in cancer[69–76]. We note that it is hard to disentangle how much of the downstream effects are caused by the specific knockdown of the target transcript or by the collateral RNA cleavage activity of LbuCas13a when targeting endogenous transcripts. Ultimately, however, both the specific knockdown of an oncogenic transcript and viability reduction due to collateral RNA cleavage are desirable effects when eliminating cancer cells.

Although we were able to deplete cells from a mixed population, we were not able to completely eliminate all target cells. With just one round of selection, we reach a similar efficiency as Shi and colleagues[51]. In addition, we showed that multiple rounds of selection are possible, reducing the proportion of target cells from 80% to just 13% of the population. There are

several potential strategies for further improvement of CRISPR-Cas13 cell selection systems. For example, the number of selection rounds could be increased, since we observed no indication of reduced efficiency with subsequent selection rounds. Furthermore, the sensitivity to collateral cleavage of cells might differ between different phases of the cell cycle, as it does for many anti-cancer drugs[77], so cell cycle synchronization may be a factor to consider as well. Finally, new Cas13 orthologs or engineered Cas13 variants with even higher collateral RNA cleavage activity than LbuCas13a could be discovered or developed through directed protein evolution approaches. We envision that CRISPR-Cas13-based cell selection systems will find broad application in both basic and clinical research, enabling the selective elimination of unedited cells, the removal of specific cell types to improve stem cell differentiation yields, and potentially the targeted elimination of cancer cells.

## Materials and methods
### Molecular cloning
For recombinant protein production, all Cas13 orthologs were cloned into a pET15b expression vector (Addgene: 62731). LbuCas13a and LshCas13a were a gift from Jennifer Doudna (Addgene: 83482 and 83487). LwaCas13a, BzCas13b, and PspCas13b were a gift from Feng Zhang (Addgene: 91909, 89898, and 103862). RspCas13d was a gift from Arbor Biotechnologies (Addgene: 108304). pET-28b-RfxCas13d-His was a gift from Ariel Bazzini & Miguel Angel Moreno-Mateos (Addgene: 141322). For the construction of the lentiviral LbuCas13a, RfxCas13d, and fluorescent protein expression vectors, all fragments were ordered from IDT as gBlocks. The fragments were cloned into the previously published CRISPR-SP-Cas9 Reporter lentiviral plasmid backbone (Addgene: 62733)[54]. All molecular cloning was done using standard molecular biology techniques using In-Fusion cloning (Takara Bio) according to the manufacturer's instruction. All plasmid sequences are available in the Supplementary Table 4. Plasmids will be made available on Addgene with the following IDs: pEF-1a_dEGFP-Y66S-IRES-PuroR-WPRE = 251210, pEF-1a_EGFP-IRES-PuroR-WPRE = 251211, pEF-1a_dEGFP-deltaFluor-IRES-PuroR-WPRE = 251212, pET15b_LbuCas13a-6xHIS = 251213, pET15b_LwaCas13a-6xHIS = 251214, pET15b_Lsh-Cas13a-6xHIS = 251215, pET15b_PspCas13b-6xHIS = 251216, pET15b_BzCas13b-6xHIS = 251217, pET15b_RspCas13d-6xHIS = 251218, pEF1-a_LbuCas13a-HA-IRES-BFP-2A-BSD = 251219, pEF-1a_RfxCas13d-HA-IRES-BFP-2A-BSD = 251220, pEF-1a_EGFP_IRES_BSD_WPRE = 251221, pEF-1a_mScarlet_IRES_BSD_WPRE = 251222.

### Cell culture
HAP1 cells were a gift from Dr. Brummelkamp (NKI Amsterdam, NL). RH30 and RD cells were a gift from Dr. Gerben Schaaf (Erasmus MC Rotterdam, NL). HEK293T cells were ordered from Takara (#632180). U2OS and ARPE19 cells were ordered from ATCC (HTB-96 and CRL-2302, respectively). All cell lines in culture were tested for mycoplasma monthly.

HAP1 cells were cultured in Iscove's Modified Dulbecco's Medium (IMDM) supplemented with 10% Fetal Bovine Serum (FBS).

HEK293T cells were cultured in Dulbecco's Modified Eagle Medium (DMEM) with 10% FBS. U2OS, RH30, and RD cells were cultured in DMEM supplemented with 10% FBS and 2 mM L-Glutamine. ARPE19 cells were cultured in DMEM/F12 with 10% FBS. All culture medium was purchased from Gibco. All cells were grown at 37 °C in a humidified atmosphere containing 5% $CO_2$.

### Lentiviral production and stable cell line generation
Lentivirus was produced in HEK293T cells. Briefly, the four packaging plasmids pHDM-Hgpm2, pHDM-Rev1b, pHDM-Tat1b, pHDM-G, and the transfer plasmid were transfected at a ratio of 2:1:1:2:4 into HEK293T cells with FuGENE HD transfection reagent (Promega, E2311) according to the manufacturer's protocol. The FuGENE to DNA ratio was 2.5:1. One day after transfection, the cell culture medium was supplemented with medium supplemented with caffeine (Sigma, C0750) and HEPES (Gibco, 15630-080), to a final concentration of 3 mM and 25 mM, respectively. The supernatant containing the lentivirus was harvested 48 h after transfection, filtered with a 0.22 µm filter (EMD Millipore, SCGP00525), and stored at 4 °C.

To generate the HAP1-dEGFP$^{Y66S}$, HAP1-dEGFP$^{\Delta fluor}$ and HAP1-EGFP lines[53], HAP1 cells were transduced with lentivirus carrying the different transgenes (Supplementary Table 4). Two days after transduction, puromycin selection was started at 0.5 µg/mL to select for integration of the transgene plasmid. After resistant cells had established themselves, the puromycin concentration was increased every 2 days to 1, 2, 5, and finally 10 µg/mL. HAP1-dEGFP$^{Y66S}$, HAP1-dEGFP$^{\Delta fluor}$ and HAP1-EGFP cells were maintained in HAP1 medium supplemented with 10 µg/mL puromycin.

To generate a RfxCas13d protein expressing HAP1-dEGFP cell line, HAP1-dEGFP cells were transduced with lentivirus carrying the pEF-1a_RfxCas13d-HA-IRES-BFP-2A-BSD plasmid (Supplementary Table 4). Two days after transduction, blasticidin selection was started at 10 µg/mL. Because the percentage of BFP positive cells remained low, cells were sorted for being BFP positive at 14 days after infection on a CytoFLEX SRT benchtop cell sorter (Beckman Coulter). Cells were maintained in HAP1 medium supplemented with 10 µg/mL blasticidin.

The RD-EGFP and RH30-mScarlet lines were generated by transducing RD and RH30 with lentivirus contain the pEF-1a_EGFP_IRES_BSD_WPRE or pEF-1a_mScarlet_IRES_BSD_WPRE transfer plasmids (Supplementary Table 4). Two days after transduction, blasticidin selection was started at 10 µg/mL. In addition, RD and RH30 cells were sorted for being EGFP or mScarlet positive at 10 days after selection started on a CytoFLEX SRT benchtop cell sorter (Beckman Coulter). Cells were maintained in culture medium supplemented with 10 µg/mL blasticidin.

### Expression and purification of recombinant Cas13 proteins
The sequences of the recombinant protein expression vectors are available in Supplementary Table 4. Expression vectors will be made available on Addgene. The different recombinant Cas13 orthologs were produced as described previously for Cas12[53]. In brief, the proteins were expressed in *Escherichia coli* BL21(DE3) and purified by Ni-NTA affinity purification followed by gel-filtration chromatography. The purified Cas13 proteins were concentrated to 75 µM using the Amicon ultra centrifugal filter MWCO 50 kDa (Merck), flash frozen in liquid nitrogen, and stored at −80 °C. Recombinant LbuCas13a protein will be shared on reasonable request.

### Preparation of guide RNAs and target RNA by in vitro transcription
Oligonucleotides carrying the T7 promoter and appropriate downstream sequence were synthesized by IDT and PCR-amplified to generate in vitro transcription (IVT) templates (see Supplementary Table 1 for template and primer sequences). The pEF-1a_dEGFP-Y66S-IRES-PuroR-WPRE plasmid (Supplementary Table 4) was used as PCR template to generate the IVT

template of the target RNA. IVT reactions were performed at 37 °C for 4 h in a buffer containing 40 mM Tris-HCl (pH 8), 20 mM $MgCl_2$, 5 mM DTT, 2 mM NTPs, 50 µg/mL T7 RNA polymerase, and 250 ng/µL DNA template. Next, to remove the DNA template, 1 µL Turbo DNase (Invitrogen: AM2238) per 100 µL IVT reaction was added and incubated at 37 °C for 30 min. The RNA was then purified using the Zymo RNA Clean & Concentrator −25 kit (R1017). Typically, four different guide RNAs were tested for each target RNA, assessing collateral cleavage by comparing total RNA degradation using the Bioanalyzer (see below). The guide RNA with the highest amount of total RNA degradation was selected for use in experiments.

### In vitro collateral cleavage activity assay
Collateral RNA cleavage detection assays were performed with IDT RNaseAlert (IDT, 11-04-02-03) according to the manufacturer's protocol. Reactions were performed in different cleavage buffers depending on the Cas13 ortholog. For LshCas13a, LwaCas13a, LbuCas13a and RfxCas13d the buffer[9] consisted of 40 mM Tris-HCl pH 7.3, 60 mM NaCl and 6 mM $MgCl_2$. For Cas13b (BzCas13b and PspCas13b), the cleavage buffer contained 10 mM Tris-HCl pH 7.5, 50 mM NaCl, and 0.5 mM $MgCl_2$[10]. For RspCas13d, the cleavage buffer used was 20 mM HEPES pH 7.1, 50 mM KCl, 5 mM $MgCl_2$ and 5% glycerol[12]. Before the RNaseAlert assay, 1 µM Cas13 and 500 nM guide RNA were preassembled in 10× cleavage buffer for 10 min at 37 °C. The RNaseAlert assays were performed using 100 nM Cas13 protein and 50 nM guide RNA, using a 50 µL reaction volume. The target RNA concentration was 0.01, 0.1, or 1 nM, as indicated in the figure. Reactions were prepared on ice, in opaque 96-well plates (Greiner, 655076). Fluorescence was measured every 5 min using a Tecan Spark plate reader, while incubating at 37 °C. Reactions were allowed to proceed for at least one hour. The collateral cleavage rate was calculated in Excel by applying the SLOPE function to the initial linear increase in fluorescence.

### Ribonucleoprotein transfection using iTOP
The Cas13 proteins and their guide RNAs were transfected into target cells using iTOP, using a protocol adapted from D'Astolfo et al.[54]. The iTOP protocol consists of four steps: (1) seeding the cells, (2) iTOP transfection mixture preparation, (3) incubating the cells in the iTOP mixture, and (4) recovery. First, cells were plated the day before transfection in Matrigel coated pates (Corning: 356231, diluted 1:100 in PBS), such that the confluency was ~70% the next day. Second, to prepare the iTOP transfection mixture, 2/5 of transfection supplement B (Opti-MEM media supplemented with 810 mM NaCl, 504 mM GABA, 1.875× concentrated N2 (Gibco: 17502048), 1.875× concentrated B27 (Gibco: 17504044), 1.875× concentrated MEM non-essential amino acids (Gibco: 11140050), 3.75 mM L-Glutamine (Gibco: 25030081), 2.5 µg/mL bFGF (Gibco: PHG0023), and 0.625 µg/mL EGF (Gibco: PHG0313)), 1/5 of Cas13 protein (75 µM) or empty protein storage buffer[53], 1/5 of guide RNA (75 µM) or nuclease-free water, and finally 1/5 nuclease-free water were combined. iTOP mixture volumes used per well were: 50 µL for 96-well plates, 150 µL for 48-well plates, 200 µL for µ-slide 8-well (ibidi, 80827) and 1000 µL for 6-well plates. Third, the culture medium was aspirated and iTOP mixture was carefully added into the wells. Next, the cells were returned to the cell culture incubator for 45 min. Fourth, immediately after incubation the iTOP mixture was aspirated and replaced by pre-warmed cell culture medium (96-well: 200 µL, 48-well: 600 µL, µ-slide 8-well: 300 µL, 6-well: 4000 µL). Finally, cells were returned to the cell culture incubator or immediately harvested for downstream analysis.

### Ribonucleoprotein transfection using electroporation
A Lonza Nucleofector 2b Device and the Cell Line Nucleofector Kit V (Lonza) were used for electroporation of HAP1-dEGFP and HEK293T cells targeting endogenous transcripts (Supplementary Figs. 5a and 6c), according to the manufactures instructions. One million cells were used per electroporation, resuspended in 100 µL Nucleofector solution V buffer containing 1 µM LbuCas13a and 1 µM guide RNA. The electroporations

were performed using the "Cell-line T-020" program. For the electroporations with RfxCas13d (Supplementary Fig. 16), HAP1-dEGFP cells were electroporated using a Bio-Rad Gene Pulser Xcell. One million cells were resuspended in Gene Pulser electroporation buffer (Bio-Rad #1652677) containing 1 μM Cas13 protein and 1 μM guide RNA in a total volume of 150 μl. Cells were electroporated in 4 mm cuvettes using a single 400 V, 5 ms square wave. After electroporation, the cells were incubated at 37 °C in 5 mL culture medium until harvest.

## Plasmid transfection

For the transient transfection of HAP1-dEGFP cells with the pEF1-a_LbuCas13a-HA-IRES-BFP-2A-BSD plasmid (Supplementary Table 4), cells were plated in 6-well plates such that the confluence was ~80% at the time of transfection. Five μg plasmid DNA was mixed with 15 μg PEI (MW 25000, Polysciences #23966) in a total volume of 500 μl opti-MEM (Gibco #31985070). The mixture was incubated at room temperature for 20 min. After incubation, the mixture was added dropwise to the culture medium in the well. Next, the 6-well plate was spun at $200 \times g$ for five minutes, after which the cells were return to the incubator.

## Total RNA isolation

Cells were collected in TRIzol reagent (Invitrogen: 15596018) and stored at −80 °C until RNA isolation. RNA for the total RNA sequencing with ERCC spike-in RNAs and CDK4 targeting experiments was isolated directly from TRIzol with the Zymo Direct-zol-96 RNA Kit (R2056), according to the manufacturer's instructions and including an in-column DNase treatment. To isolate the total RNA for all other experiments, chloroform was first added to the TRIzol at a ratio of 1:5 and samples were mixed by briefly vortexing. Next, samples were centrifuged for 15 min at $12,000 \times g$ at 4 °C. Total RNA was isolated from the colorless upper aqueous phase using the Zymo RNA Clean & Concentrator −5 kit, including an in-column DNase treatment, according to the manufacturer's instructions.

## Bioanalyzer and Femto Pulse

Total RNA from the experiments targeting CDK4 in RD and RH30 was analyzed on the Agilent Femto Pulse according to the manufacturer's instructions. Total RNA integrity of all other samples was analyzed on the Bioanalyzer using either the RNA nano or RNA pico chips, depending on the available RNA amount. The area under the curve (AUC) was calculated in R with the "AUC" function from the DescTools package using the default settings[78]. To account for differences in the amount of RNA loaded, all samples were normalized to their total AUC. This was done by dividing the sample fluorescence values by the sample total AUC and multiplying by the mean AUC across all samples. In our hands, this method resulted in a more accurate normalization than using the sample concentration estimates supplied by the Bioanalyzer software. In Fig. 2a, b, one replicate of *18S rRNA* targeting in U2OS, *GAPDH* targeting in U2OS, and non-targeting in ARPE19 was excluded due to an unstable baseline, extremely low signal, and signal dropout halfway though the run, respectively. These samples were included in Fig. 2c and fit with the rest of the data.

## RT-qPCR and 5′−3′ RT-qPCR

RNP transfection by iTOP was performed in 96-well plates, as described above. Besides Cas13 protein and either a targeting or a non-targeting guide RNA, an empty transfection control was always included. All ddCq values are relative to this empty transfection control. Total RNA was isolated and DNase-treated as described above. An anchored oligo(dT) primer was used for reverse transcription using SuperScript III reverse transcriptase (Invitrogen, 18080093), according to the manufacturer's protocol. qPCR was performed in triplicate using Bio-Rad iQ SYBR Green Supermix with 300 nM primers. The primer sequences used for reverse transcription and qPCR can be found in Supplementary Table 2.

Traditionally, the ratio between the Cq values of the 3′ and 5′ primer pair has been used to quantify transcript integrity[55]. However, this will result in different ratios for the same amount of degradation when expression levels are different ($\frac{20}{25} \neq \frac{25}{30}$). Therefore, to quantify the transcript integrity, we used a modified version of the ddCq method where the 3′ primer pair takes on the function of the housekeeping gene. First, to calculate the 5′−3′ dCq values, the 3′ Cq value was subtracted from the 5′ Cq value for each sample. Next, to calculate the 5′−3′ ddCq, the mean 5′−3′ dCq value of the empty transfection controls was subtracted from the 5′−3′ dCq values of the treatments. To avoid confusion with the method were the ratio between the 3′ and 5′ primer pair is used, we called the assay 5′−3′ RT-qPCR here instead.

## Western blot

First, the cells were put on ice and washed three times with ice cold PBS. Cells were collected in ice cold RIPA buffer and kept on ice for 20 min. The lysate was spun down at $16,000 \times g$ at 4 °C for 20 min. The supernatant was transferred to a new tube and the protein concentration was measured using a BCA assay (Thermo Scientific, 23225). Before loading on gel, samples were mixed with Laemmli buffer (Bio-Rad, 1610747) and heat denatured at 95 °C for 5 min. Twenty μg protein lysate was loaded per well, unless otherwise indicated. Samples were run on 4–20% Mini-PROTEAN precast gels (Bio-Rad, 4568094) at 140 V until bromophenol blue reached the bottom of the gel. The protein was then transferred to a PVDF membrane using a wet transfer at 300 mA for 90 min. After transfer, the membrane was blocked in blocking buffer (PBS with 5% w/v skim milk powder, 1% w/v BSA, and 0.1% Tween 20) at room temperature for 20 min. Next, the membrane was incubated overnight in blocking buffer with an anti-HA tag (ThermoFisher, 26183) and a GAPDH loading control antibody (ThermoFisher, MA5-15738), diluted at 10000× and 1000×, respectively. Next, the membrane was washed three times for five minutes in PBST (PBS with 0.1% Tween 20). The membrane was incubated for two hours at room temperature with Goat Anti-Mouse IgG H&L (HRP) (Abcam, ab6789), diluted 1:1000 in blocking buffer. After incubating with the secondary antibody, the membrane was again washed three times for five minutes in PBST. Finally, ECL substrate (ThermoFisher, 32106) was applied to the membrane and the signal was detected on a Bio-Rad GelDoc Go System.

## Nanopore cDNA sequencing

The experiment and subsequent library preparation consisted of seven steps: (1) RNP transfection, (2) RNA isolation, (3) rRNA depletion, (4) poly(A) tailing, (5) reverse transcription, (6) second strand synthesis, and (7) barcode and sequencing adapter ligation. Manufacturer's instructions were followed for all kits unless otherwise noted.

To ensure enough RNA for sequencing, this experiment was performed in 6-well plates, plating $6 \times 10^5$ HAP1-dEGFP$^{Y66S}$ cells per well the day before the transfections. One well was reserved as a no transfection control. iTOP was performed as described above, using an iTOP mixture volume of 1 mL per well. Each timepoint had three conditions: an empty transfection control, a transfection with LbuCas13a and a non-targeting guide RNA, and a transfection with LbuCas13a and a dEGFP targeting guide RNA (SP2). Cells were collected in TRIzol at 0, 50, 200, 500, and 1440 min after iTOP. Total RNA was isolated and DNase-treated as described above. RNA integrity was assessed on the Bioanalyzer (Supplementary Fig. 10A). Next, 5 μg total RNA of each sample was used for rRNA depletion with the RiboMinus Eukaryotic system v2 (Invitrogen, A15026). rRNA depletion was confirmed with the Bioanalyzer (Supplementary Fig. 10B). In order to also detect cleaved RNA fragments without a poly-A tail, we polyadenylated the rRNA depleted RNA with 0.25 units/μL *E. coli* Poly(A) polymerase (NEB, M0276L) in the presence of 1 mM ATP and 1 unit/μL RNasin® (Promega, N2515) for 30 min at 37 °C. Afterwards, the RNA was cleaned with the Zymo RNA Clean & Concentrator −5 Kit (R1013). Next, reverse transcription was performed using SuperScript IV RT (Invitrogen, 18090010) and anchored oligo dT$_{20}$ primers (Supplementary Table 2). The reverse transcription reaction was incubated at 50 °C for 15 min followed by 80 °C for 10 min. To remove RNA before second strand synthesis, 1 μL of RNase Cocktail (Thermo, AM2286) was added after reverse transcription and incubated at 37 °C for 20 min. The NEBNext second strand synthesis kit

(NEB, E6111) with 100 ng random hexamer primers (Promega, C1181) per sample was used for second strand synthesis. Next, dsDNA was purified using AMPure XP beads (Beckman Coulter: A63882) at a ratio of 2 beads to 1 sample. The NEBNext Ultra II End Repair/dA-tailing Module (NEB, E7546S) was used before ligating Nanopore native barcodes (EXP-NBD104&114). Finally, samples were pooled at equal mass and the sequencing adapter (SQK-LSK109) was ligated. 1000 ng DNA was loaded on the Promethion flow cell (FLO-PRO002) and sequenced for 3 days. Three independent replicates were performed.

### Nanopore data analysis

During sequencing, the Nanopore MinKNOW software (version 21.11.7, Guppy version 5.1.13) called bases in real-time by the using the "Super-accurate" base calling setting and demultiplexed samples. Barcodes were trimmed either by the MinKNOW software or with Porechop (https://github.com/rrwick/Porechop). For replicate 1, the 0 min after transfection, dEGFP targeting sample had a very low DNA yield after library preparation and consequently much fewer reads than the other samples. Therefore, it was excluded from all analysis. Reads were mapped to the transcriptome using minimap2 2.21-r1071[79]. Transcriptome alignment was done using the *-ax map-ont* arguments, against a merge of the Ensemble Homo_sapiens. GRCh38.cdna.all (release 104), Ensemble Homo_sapiens.GRCh38.ncrna (release 104), the entire pEF-1a_dEGFP-Y66S-IRES-PuroR-WPRE plasmid (Supplementary Table 4) and the 18S and 28S rRNA. Mapping rates were similar across samples, with dEGFP targeting samples showing a small decrease at 500 and 1440 min after transfection (Supplementary Fig. 12a). Only the primary alignment was used for analysis. To generate the density plots of reads mapped to protein coding transcripts in Fig. 6b and Supplementary Fig. 12c, all histone transcripts and WDR74 were excluded. Histone transcripts and WDR74 were so abundant that they obscured all other mRNAs in the density plot. The extremely high abundance of WDR74 was caused by a small nuclear RNA mapping to WDR74-204. All analysis was performed in RStudio, using data.table[80] and ShortRead[81] for data loading, Tidyverse[82] for data handling and ggplot2[83] for plotting. Raw sequencing reads are available online at https://www.ncbi.nlm.nih.gov/sra/PRJNA912090. The scripts used for calculating transcript coverages and plot generation is available on https://github.com/Geijsenlab/cas13.

### RNA sequencing and analysis

For the RNA-seq experiment, HAP1-dEGFP$^{Y66S}$ cells were transfected using iTOP, as described above. The day before transfection, $6 \times 10^5$ cells per well had been plated in 6-well plates. Treatments were: an empty transfection, a transfection with LbuCas13a but no guide RNA, a transfection with Lbu-Cas13a and a non-targeting guide RNA, and finally a transfection with LbuCas13a and a dEGFP targeting guide RNA (SP2). All treatments were done in triplicates. After 16 h, cells were lysed in TRIzol reagent. Total RNA was isolated and DNase-treated as described above. mRNA was then extracted using the NEBNext Poly(A) mRNA Magnetic Isolation Module (NEB, E7490S) and RNA-seq libraries were prepared using a NEBNext Ultra Directional RNA Library Prep Kit for Illumina (NEB, E7420L). RNA-seq libraries were sequenced on an Illumina NextSeq500 Instrument with at least 10 million reads per library. Quality control by FastQC[84] and RSeQC[85] did not reveal any major issues. An index was generated using the RefSeq GRCh37 assembly and reads were aligned and quantified using STAR[62] and featureCounts[86]. Differentially expressed genes were determined using DESeq2[87]. A design formula was used with the variables "protein", "guide", and "targeting". The "protein" variable indicates whether a sample had been transfected with recombinant LbuCas13a protein (all but the empty iTOP condition), the "guide" variable indicates whether a sample had been transfected with a guide RNA (both the non-targeting and targeting condition), and the "targeting" variable indicates whether a sample had been transfected with both LbuCas13a and a dEGFP targeting guide RNA. For each variable of the design formula, genes with an adjusted *p*-value < 0.05 and log$_2$(fold change) > |1| were regarded as differentially expressed genes, and they were subjected to subsequent enrichment analysis. GO term and

pathway enrichment analysis was done using ShinyGO[88] V0.82, supplying all detected transcripts as background. RNA-seq data is available online at the NCBI Gene Expression Omnibus (GSE220759). The code of the differential gene expression analysis is available on https://github.com/Geijsenlab/cas13.

### Total RNA sequencing with ERCC spike-in control RNAs

HAP1-dEGFP cells were plated in 96-well plates and transfected by iTOP as described above. Treatment conditions were: untreated cells, empty iTOP transfections, iTOP with LbuCas13a and a non-targeting guide RNA, or iTOP with LbuCas13a and a dEGFP targeting guide RNA (SP2). Cells were collected in TRIzol at 50, 200, and 1440 min after iTOP. Three replicates were performed. RNA was isolated using the Zymo Direct-zol 96 RNA kit (R2056), including the in-column DNase treatment. RNA integrity and collateral RNA cleavage in the dEGFP targeting condition was confirmed using the Agilent Femto Pulse system. Next, 2 μl of 1:1000 diluted ERCC RNA Spike-In Mix 1 (ThermoFisher #4456740) was added to 100 ng total RNA of each sample. Samples were submitted to Macrogen for library preparation and total RNA sequencing. The Watchmaker RNA library prep kit with Polaris Depletion - rRNA/Globin (HMR) was used for library preparation. Samples were sequenced on NovaSeq X Plus using 101 bp paired-end sequencing, yielding >30 million paired reads per sample.

Adapters were trimmed from the reads with TrimGalore 0.6.6. Next, STAR 2.7.11b[62] was used to align reads to the combined GRCh38 human genome, dEGFP-PuroR construct, and the ERCC spike in RNA sequences. A count table was generated with featureCounts version 2.0.3[86] using the -M -O –fraction flags. Replicate three of the empty iTOP condition at the 50 min after iTOP timepoint was excluded from analysis because it failed library prep QC, had a high percentage of rRNA mapped reads, and low correlation between ERCC spike-in RNA counts and their concentrations. Differential gene expression analysis was performed with DESeq2[87]. In short, a separate DESeq model was constructed for each timepoint. The same design formula was used for each timepoint, containing three variables: "iTOP", "RNP", and "Targeting". These variables indicate whether samples had been exposed to the iTOP transfection procedure (all but the untreated samples), whether samples were transduced with LbuCas13a and any guide RNA (both the non-targeting and targeting conditions), and whether they were transduced with LbuCas13a and a targeting guide RNA, respectively. The ERCC spike-in control RNAs were used to estimate size factors before running DESeq() on the model. The code of the differential gene expression analysis is available on https://github.com/Geijsenlab/cas13. Functional enrichment analysis was performed using string-db.org with default settings[89]. Source data is available at the NCBI GEO (GSE308909).

### Live cell imaging

The live cell imaging protocol used here was adapted from Wallberg et al.[60]. HAP1 and HAP1-dEGFP$^{Y66S}$ cells were imaged in imaging medium (FluoroBrite™ DMEM, 10% FBS, 2.5 mM CaCl$_2$) supplemented with 2 μL CellTox™ Green (Promega, G8741) and 5 μL Annexin V-AV647 (Invitrogen, A23204) per milliliter. Cells were seeded in 8-well μ-Slides (ibidi, 80827) coated with a 1:100 dilution of Matrigel (Corning, 356231), plating 55,000 cells per well. The next day, iTOP was performed as described above, using an iTOP mixture volume of 200 μL per well. After removing the iTOP buffer, wells were washed with 300 μL PBS before adding 300 μL imaging medium. This extra PBS wash step prevents protein or salt precipitation during imaging. After iTOP, cells were allowed to recover in the cell culture incubator for up to four hours. Next, cells were moved to the Leica AF7000 and allowed to equilibrate for one hour (37 °C, 5% CO$_2$). During live imaging, pictures were taken every 15 min at four positions per well for 20 h, using a 20× Dry N.A. 0.50 objective. Finally, the amount of Annexin V and CellTox™ Green positive cells were counted with CellProfiler[90] using the IdentifyPrimaryObjects module. Counts from the four imaged positions per well were summed and treated as one replicate. One replicate of the CTRL, SP1, and SP2 treatments in HAP1 was excluded due to having a 2 to 3 times higher starting number of annexin V positive cells.

## CellTiter-Glo cell viability assay

RD and RH30 were plated in white-walled clear bottom 96-well plates. RD was seeded at a density of 15,000 cells per well, RH30 at 17,000 cell per well. Three wells were seeded per condition, acting as technical replicates. The next day, they were transfected by iTOP as described above. The CellTiter-Glo assay (Promega, G7571) was performed at 2 days after the iTOP transfection, following the manufacturer's instructions. The luminescence was measured with the GloMax Explorer, using a 0.3 s integration time. Technical triplicates were averaged and the assay was repeated three times.

## Cell selection experiments

For the cell selection experiments, HAP1-EGFP cells were mixed with HAP1 cells at a ratio of roughly 4:1. HAP1-EGFP were selected against by introducing LbuCas13a with an EGPF targeting guide RNA (SP2) using iTOP. The percentage of EGFP positive cells was measured by FACS on the BD FACSCanto II (BD Bioscience), measuring at least 10,000 cells per sample. To generate HAP1-EGFP lines with different levels of *EGFP* expression, cells were sorted into fifteen different populations with increasing EGFP intensities using the BD FACSJazz cell sorter (BD Bioscience). We sorted the cells in complete cell culture medium. After sorting, the three populations with the lowest EGFP intensity were cultured in medium without puromycin. The next three lowest populations were cultured at 2 µg/mL puromycin, the following groups of three at 5 µg/mL, 10 µg/mL and finally 20 µg/mL. For the selection experiment, the different EGFP lines were mixed with wildtype HAP1 cells and subjected to selection as described above. Additionally, in parallel they were also plated in a separate well. These parallel wells were harvested in TRIzol immediately after the mixed population had been subjected to iTOP, and used to determine their EGFP expression levels. Conventional RT-qPCR was performed as described above for $5'-3'$ RT-qPCR. qPCR primers can be found in Supplementary Table 2.

To repair the dEGFP$^{\Delta fluor}$ mutation, cells were transfected with recombinant AsCas12a protein, guide RNA (Supplementary Table 3), and a LAHR repair template (Supplementary Table 3) using iTOP. For each well of a 96-well plate, 50 µL of iTOP mixture that contained 20 µL of transfection supplement B, 10 µL of AsCas12a protein (150 µM), 10 µL of guide RNA (300 µM), and 10 µL of 500 pmol repair template in nuclease-free water was used. Three days after iTOP transfection, the cells that had not been edited were selected against by delivering LbuCas13a and a dEGFP$^{\Delta fluor}$ targeting guide RNA (Supplementary Table 1) using iTOP. Finally, the percentage of EGFP positive cells was assessed on the BD FACSCanto II three days after selection with LbuCas13a, measuring at least 10,000 cells per sample.

To select against either RH30-mScarlet or RD-EGFP cells we first mixed them aiming for a 2:1 ratio. The cell mix was subjected to two rounds of treatment using iTOP delivery, with four days between the two treatment rounds. The treatment conditions were empty iTOP, iTOP with LbuCas13a and a non-targeting guide RNA, iTOP with LbuCas13a and a *CDK4* targeting guide RNA (SP1), or iTOP with LbuCas13a and a *dEGFP* targeting guide RNA. The Cas13 protein and guide RNA concentration was 15 µM. The guide RNAs used for these experiments were produced by IDT (Supplementary Table 1). The percentage of RH30-mScarlet and RD-EGFP cells was assessed by FACS three days after each round. An example of the gating strategy used for assessing the percentage of fluorophore positive cells is shown in Supplementary Fig. 17.

## Statistics and reproducibility

All analysis was performed in R, using the Tidyverse collection of packages for data loading, handling and plotting[82]. Welch's *t*-tests were performed using the stat_compare_means function from the ggpubr package[91]. The number of AnnexinV and CellTox Green positive cells in the live imaging time course (Fig. 3b) were compared using a repeated-measure ANOVA with the formula: num_positive_cells ~ time + treatment + cell_line + Error(sample_id/time). DESeq2 was used for differential gene expression analysis, the code of which is available on GitHub (https://github.com/Geijsenlab/cas13). Except for the live imaging (Fig. 3b) and multiple cell selection rounds (Fig. 4c and f), all measurements were on distinct samples.

Data was tested for normality by Q–Q plot. All performed statistical tests were two-sided. P-values below 0.05 were considered statistically significant.

## Reporting summary

Further information on research design is available in the Nature Portfolio Reporting Summary linked to this article.

## Data availability

The RNA-seq data has been deposited in the NCBI Gene Expression Omnibus (GSE220759, https://www.ncbi.nlm.nih.gov/geo/query/acc.cgi?acc=GSE220759). The total RNA-seq data with ERCC spike-in RNAs has been deposited at the NCBI Gene Expression Omnibus (GSE308909, https://www.ncbi.nlm.nih.gov/geo/query/acc.cgi?acc=GSE308909). The Nanopore sequencing data has been deposited at the NCBI Sequence Read Archive (PRJNA912090, https://www.ncbi.nlm.nih.gov/bioproject/912090). Western blot source data is shown in Supplementary Fig. 18. All other source data is available in Supplementary Data 1 (main figures) and 2 (Supplementary Figures). Plasmids will be deposited to Addgene (IDs 251210-251222).

## Code availability

Scripts for the RNA-seq analysis, total RNA-seq analysis and Nanopore sequencing coverage calculation and analysis and are available on GitHub (https://github.com/Geijsenlab/cas13). TrimGalore, which we used for RNA-seq read trimming and QC, can be found on GitHub (https://github.com/FelixKrueger/TrimGalore). PoreChop, used for nanopore adapter trimming, is available on GitHub (https://github.com/rrwick/Porechop).

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

## Acknowledgements

The authors thank Stefan van der Elst of the Hubrecht Institute FACS facility, the LUMC FACS facility, and the Utrecht sequencing facility (USEQ) for technical help. The Novo Nordisk Foundation Center for Stem Cell Medicine (reNEW) is supported by a Novo Nordisk Foundation grant number *NNF21CC0073729*. This research is supported by the Dutch Technology Foundation *STW* (15804), which is part of the Netherlands Organisation for Scientific Research (NWO), and which is partly funded by the Ministry of Economic Affairs. Dit project is ondersteund door het Prinses Beatrix Spierfonds onder projectnummer W.OR18-11. Zhihan Zhao received a scholarship (201706890022) from the China Scholarship Council. Mengyuan Li received a scholarship (202206300033) from the China Scholarship Council.

## Author contributions

J.F.B. experimental design, execution, and help writing the manuscript. Z.Z. experimental design, execution, and help writing the manuscript. M.L. technical assistance. D.K. technical assistance. P.S. experimental design and help writing the manuscript. N.G. conceptual and experimental design, project supervision, and help writing the manuscript.

## Competing interests

The authors declare the following competing interests: N.G. is co-founder of NTrans Technologies and Divvly. No conflicting interest.
