## [Transparent Peer Review file · Communications Biology]

Temporal dynamics of collateral RNA cleavage by LbuCas13a in human cells

Corresponding Author: Dr Niels Geijsen

Version 0:

Reviewer comments:

Reviewer #1

(Remarks to the Author)

The study provides new insights into Cas13 activity in human cells, particularly its kinetics, cleavage preferences, and cellular responses. The Nanopore sequencing approach adds a mechanistic dimension, which could be useful for future Cas13 engineering. If rigorously validated, this could be useful for designing RNA-targeting CRISPR applications where collateral activity is harnessed (e.g., for cell elimination).

However, multiple claims require stronger validation, especially the unbiased nature of cleavage (uracil-containing loops) and transcriptome-wide RNA degradation. Is cell apoptosis truly due to Cas13 activity, or are other stress pathways involved? The claim of "major alterations to the total RNA profile" needs better quantification—does this include widespread transcript degradation or just a few highly expressed RNAs (e.g., GAPDH, RPS19, 18S rRNA)?

Also, control selection (e.g. RNA seq data) and experimental design issues may weaken their conclusions. Particularly, their RfxCas13d recombinant seems not functional, making comparisons among Cas13 orthologs potentially misleading.

Minor concerns:

1. Suppl. Figure 10: The term transduction/empty transduction should be replaced with transfection for accuracy.
2. Line 324: The term dEGFP-IRIS-PuroR appears to contain a typographical or conceptual error. Do the authors mean IRES (internal ribosome entry site) instead of IRIS? If so, this should be corrected for accuracy.

For the transcript coverage plots, why is the No iTOP condition (blue line) only shown at the 0-minute timepoint in Figure 6B, Figure 6C, and Supplementary Figure 12,13,14? To fully assess the impact of iTOP treatment over time, the No iTOP condition should also be included at later timepoints for proper comparison. If this data is unavailable, the authors should clarify why it was omitted.

3. Lines 233 onwards: Ensure consistency in terminology—EGFP, dEGFP, or GFP. If these refer to distinct GFP variant sequences, please clarify; otherwise, standardize the naming throughout the text.

Major concerns:

1. Suppl. Figure 1-RfxCas13d Collateral Activity:

RfxCas13d did not show any collateral activity in this assay. Why? Cas13d is one of the most frequently reported orthologs exhibiting collateral activity. Control experiments are needed to determine whether the recombinant proteins are functional and whether the gRNAs (how many were used per ortholog?) are effective in achieving comparable target activity. To confirm on-target activity, in vitro cleavage assays against dEGFP and PAGE gel analysis of cleaved dEGFP RNA should be performed. Additionally, non-targeting control gRNAs are necessary to establish baseline activity.

2. Line 120 – Supplementary Figures 2 and 3 (Comparison of Orthologs to LbuCas13a in Human Cells):

The comparison of different orthologs in Figure 1 and Supplementary Figures 2 and 3 is not entirely fair, as the delivery methods and controls differ between experiments.

- Figure 1A: HAP1-dEGFP data are overlaid with the negative control HAP1 cells.
- Supplementary Figure 2: The overlay includes a non-targeting gRNA line with targeting gRNAs, but the specific cell line (HAP1-dEGFP or HAP1) is not indicated.
- Supplementary Figure 3: The authors used electroporation instead of iTOP (as used in Figure 1A and Supplementary Figure 2). Why was this approach chosen? Additionally, the control condition in Supplementary Figure 3 is unclear—there is no indication of HAP1-dEGFP or a non-targeting gRNA.

3. Lines 138–145 – Supplementary Figure 5 and Figure 1C:

To ensure a valid comparison of ortholog activities, the authors should use consistent controls. Specifically:

- a. A non-targeting gRNA should be included in Figure 1C.
- b. Or, HAP1 cells should be included in Supplementary Figure 5.

4. Lines 151–157, 387–388: The data do not support their conclusion that LbuCas13a exhibited the highest collateral RNA cleavage activity both in vitro and in cells. We lack strong evidence for collateral cleavage with these orthologs (they refer to RfxCas13d here, Ai et al., 2022; Y. Li et al., 2023; Shi et al., 2023). "...or RfxCas13d (Ai et al., 2022; Y. Li et al., 2023; Shi et al., 2023), we did not find collateral RNA cleavage activity in human cells with these orthologs."

The recombinant RfxCas13d they generated does not appear functional, which likely explains why their results contradict findings from other groups. To claim that a Cas13 variant lacks collateral activity, it must first be demonstrably active. Therefore, RfxCas13d should be excluded from the comparison after Figure 1, and any claims regarding its lack of collateral activity should be removed. Instead of arguing that LbuCas13a exhibits stronger collateral activity than RfxCas13d, they should focus on comparisons among functionally validated enzymes.

Line 158 – Supplementary Note 1:

The discussion in Supplementary Note 1 provides a more balanced interpretation of the data, acknowledging that the recombinant RfxCas13d may not have been functional in their in vitro assays. Their statement—"In any case, we cannot draw any conclusions about the in vitro activity of RfxCas13d."—should be included in the main text (e.g., at Line 108, Suppl. Figure 1) to ensure transparency and prevent overinterpretation of the results.

- Revise the main text to remove misleading claims about RfxCas13d.
- State that their recombinant RfxCas13d might not have been functional, preventing meaningful comparisons.
- Incorporate key points from Suppl. Note 1 into the main text for clarity.

5. Line 290, Figure 5A: What is the control used in this volcano plot? Please clarify which condition was used for normalization and specify the negative control.

6. Line 278: Before gaining insight into the cellular response to collateral RNA cleavage, the authors need to first assess the extent of global RNA degradation caused by collateral activity. To do so, they should include an earlier time point (50 min or 200 min) in their RNA-seq analysis. This would help determine whether widespread RNA degradation occurs shortly after Cas13 activation.

7. Lines 288–290, 671–673: The authors claim "substantial changes in the transcriptome. Indeed, we found 721 significantly upregulated and 36 significantly downregulated genes in response to LbuCas13a activation." However, the Methods section states that four conditions were used:

- Empty transfection
- LbuCas13a without guide RNA
- LbuCas13a with a non-targeting guide RNA
- LbuCas13a with a dEGFP-targeting guide RNA (SP2)

Despite having three negative controls, Figure 5 presents only a single volcano plot and one GO analysis. It is unclear which control was used for normalization in these analyses. Was Cas13 + targeting gRNA normalized to an empty transfection, Cas13 alone, Cas13 + non-targeting gRNA? This must be explicitly stated in both the main text and the figure legend. Additionally, the authors should provide volcano plots comparing Cas13 + dEGFP gRNA against the other two controls. Based on my experience with RNA-seq of CRISPR-expressing cells, the volcano plot in Figure 5 appears to be normalized against an empty transfection. If this is the case, the comparison is inappropriate, as overexpression of a bacterial protein alone can trigger an innate immune response—independent of Cas13 effector activity. Even a catalytically dead Cas13 can induce a similar effect.

Without a clear statement on the chosen control for Figure 5 and without presenting plots for the other two negative controls, I am concerned that the authors have selectively presented their data.

8. Figure 5A: Importantly, the authors need to indicate which dot represents dEGFP, the intended Cas13 activation and target.

The volcano plot appears unrefined and requires improvement for publication. The figure legend resembles the default settings of EnhanceVolcano and should be revised to clearly define categories such as not significant, upregulated, and downregulated genes. The current labels (Log2FC, p-value, p-value and log2FC) are cutoffs and should be corrected for better interpretability.

Additionally, the not significant gene group is missing from the plot, yet the legend includes 'Sig', which needs clarification. It is also unclear whether blue dots (in the legend) are present in the plot. Lastly, given that the dataset contains 22,465 genes, the dot size is too large, reducing clarity. The size should be adjusted to improve readability and ensure most significant individual data points are distinguishable.

9. Lines 316–318: The statement "The abundance and read length of reads mapped to protein-coding transcripts greatly decreased at 50 and 200 minutes after transfection (Figure 6A and Supplementary Figure 11B)" is not fully supported by the data.

In Figure 6A, the density plots appear lighter in color, and the center of the distribution shifts toward the bottom-right corner, indicating both a reduction in read abundance and a shortening of read length. However, in Supplementary Figure 11B, while the color intensity decreases (suggesting fewer reads), the center of the density plot does not shift. This implies a

reduction in read abundance but no clear evidence of a change in read length.

Additionally, the 0-minute timepoint data is missing in Supplementary Figure 11B, making it difficult to determine whether there was already a lower starting abundance. To strengthen the conclusions, the authors should provide the missing timepoint data and include at least one more biological replicate to confirm reproducibility.

10. Lines 342–344: "Transcript coverage dips often map to predicted stem-loop structures, with uracil-containing single-stranded loops on the 5' side of the coverage dip, suggesting that the 5' side of the dip is the actual cleavage location". This statement lacks sufficient direct evidence. While the correlation between cleavage sites and predicted secondary structures is intriguing, the data do not conclusively demonstrate that the 5' side of the dip is the primary cleavage site. Additional validation, such as structural probing or mutagenesis experiments (on U), would be required to confirm this. Furthermore, transcript degradation and secondary structure mapping in high-throughput sequencing can be influenced by biases unrelated to direct Cas13 activity. The authors should provide more substantial experimental support before making this conclusion.

11. Line 350–352, 358: The statement "showing a preference for uracil-containing single-stranded loop structures" should be framed as a proposed model rather than a definitive conclusion. The current data suggest a correlation but do not establish causation. Additional experimental validation, such as structural probing or mutagenesis, would be necessary to confirm this preference.

12. Line 349-350, Line 357: "LbuCas13a and suggests that 350 both mRNAs and cytoplasmic ncRNAs are cleaved without bias." "All cytoplasmic RNAs seems to be cleaved in an unbiased manner."

This claim is not well supported by their data. Their evidence is based on reduced or shorter reads observed in nanopore sequencing, but they only closely analyze a few countable RNAs (RPS19, GAPDH, and 18S rRNA). To accurately assess the extent of Cas13-mediated RNA cleavage, the authors should include an earlier RNA-seq time point (e.g., 50 min or 200 min) to capture initial transcript downregulation and compile a comprehensive list of all cleaved transcripts aligned to the human reference genome.

Additionally, the claim of unbiased cleavage overlooks spatial constraints on RNA accessibility. The cytoplasm contains structured, membrane-less organelles such as stress granules and P-bodies, which can sequester specific RNAs and limit Cas13 access. Certain ncRNAs also localize to membrane-bound organelles, further restricting their availability for cleavage. Without demonstrating that all cytoplasmic RNAs are equally accessible, the claim of unbiased cleavage is premature and should be revised to account for these biological constraints.

13. Line 390-393: This explanation is unclear. Are they suggesting that Cas13 degrades its own mRNA when expressed as a transgene, thereby reducing its protein levels? Or do they mean that transgene-expressed Cas13 is somehow less active than RNP-delivered Cas13? If guide RNA-independent cleavage is limiting expression, do they have data showing lower Cas13 protein levels in transgene-expressed cells compared to RNP delivery?

Reviewer #2

(Remarks to the Author)

This study by Bot et al. investigates whether CRISPR-Cas13 enzymes – which target RNA – exhibit the same collateral RNA cleavage in human cells as seen in bacteria and in vitro. The authors compare six Cas13 orthologs and find LbuCas13a uniquely capable of robust collateral RNA cutting in human cell lines when delivered as a ribonucleoprotein (RNP). Upon targeting a specific RNA (e.g., a GFP reporter), LbuCas13a rapidly cleaves not only the target but also other RNAs, causing a characteristic degradation pattern in total RNA within 50–100 minutes. This widespread RNA degradation triggers cellular stress responses and upregulation of innate immune genes, ultimately leading to apoptotic cell death by ~15–24 hours post-transfection. The team exploits this phenomenon as a cell elimination tool: cells expressing a target RNA can be selectively killed, allowing depletion of those cells from a mixed population. They demonstrate repeated rounds of Cas13a RNP delivery can progressively enrich or deplete cells based on expression of a target transcript. Finally, using Nanopore long-read sequencing, the authors map the precise temporal dynamics and cleavage sites of collateral damage. They show that LbuCas13a cleaves cytoplasmic mRNAs globally and without sequence bias (most transcripts are cut into short fragments), with a preference for cutting at uracil-rich loop structures in RNA, consistent with in vitro studies. Overall, the work provides a comprehensive picture of when, where, and how LbuCas13a causes collateral RNA cleavage in human cells, the resulting cellular responses, and a proof-of-concept application for selective cell ablation.

Collateral (bystander) RNA cleavage by Cas13 has been well-documented in bacteria and test tubes, but its occurrence in mammalian cells has been controversial and context-dependent. This study is novel in definitively showing that one particular Cas13 enzyme, LbuCas13a, can indeed trigger robust collateral RNA degradation in human cells under certain delivery conditions. Bot et al. not only identify LbuCas13a as an outlier with strong collateral activity, but also detail the kinetics (onset within <1 hour) and sequence preferences of cleavage inside cells – which to my knowledge has not been mapped at nucleotide resolution before.

In summary, the manuscript addresses a timely question in CRISPR biology, provides compelling evidence of a unique Cas13 behavior in human cells, and opens a new avenue for using CRISPR-Cas13 as a research and selection tool. These contributions are significant for the field of RNA-targeting CRISPR systems.

Please see detailed comments and suggestions below:

1. The authors convincingly show LbuCas13a has strong collateral RNase activity in human cells when delivered as an RNP, but virtually no activity when expressed via plasmid. This discrepancy is important. A major point to address is why expression fails – the manuscript hints at guide-independent activity or insufficient protein levels. To bolster this, the authors could consider additional evidence or discussion. For instance, was any collateral RNA degradation detectable at later times or higher plasmid doses? Did the cells expressing Cas13a show any growth disadvantage (even if bulk RNA looked normal)? While not strictly necessary, exploring a lentiviral or inducible expression system might clarify if constant low Cas13a expression allows cells to adapt or if the enzyme is inherently self-limited. At minimum, expanding the discussion in Supplementary Note 1 (or main text) on why only RNP delivery works would be valuable. This will guide others in the field: it suggests that to observe collateral effects, transient high-level delivery is required, perhaps because cells otherwise quench the activity.

2. The authors report collateral cleavage in all tested cell lines (HAP1, RD, U2OS, ARPE19) with no differences in the RNA fragment pattern (same fragments appear). Interestingly, HAP1 had a quantitatively greater response. A major question is: what causes the difference in magnitude? The authors propose higher RNP uptake in HAP1 (since iTOP was optimized there). This is plausible, but could there be biological differences? For example, do some cells have higher RNase tolerance or RNA turnover that affects fragment detection? Or differences in innate immune priming that might degrade Cas13 sooner? While the current data suffice to say it's not cell-line specific qualitatively, the manuscript might acknowledge that efficiency can vary.

3. The study effectively demonstrates that LbuCas13a's ability to deplete target cells correlates with target RNA expression levels (Figure 4E), a crucial finding with significant implications for practical application of this cell selection system. However, this aspect warrants more detailed characterization to enhance the predictive power and utility of the approach. While the dCq values provide a relative measure of expression, establishing a quantitative minimum threshold of target RNA abundance required for effective cell elimination would make the findings more broadly applicable. Converting the relative measurements to absolute transcript copy numbers per cell (via digital PCR or RNA spike-in standards), determining at what expression level the depletion efficiency becomes suboptimal, and testing a broader panel of endogenous transcripts with varied abundance levels would provide researchers with clear guidelines for selecting appropriate target RNAs.

4. The manuscript mentions that the HAP1 cells used are “mostly diploid,” despite this line's description as near-haploid. Because ploidy status can significantly affect gene dosage, RNA expression levels, and susceptibility to CRISPR-based manipulations, it would strengthen the manuscript to thoroughly characterize the ploidy of the HAP1 population used in these experiments.

5. The manuscript would benefit from a more detailed discussion of how the findings compare with previous studies showing limited collateral cleavage in eukaryotic cells.

Reviewer #3

(Remarks to the Author)

Bot et al. investigated the mechanisms of Cas13-mediated collateral activity across multiple mammalian cell lines and proposed its potential application for targeted cell elimination. The application would be of high potential if the method could be used for endogenous transcripts, such as oncogenes. The study is generally well designed, and the data provide valuable insights into the temporal kinetics of Cas13-induced RNA degradation. However, some measurements obtained from different cell lines or samples need to be normalized based on delivery efficiency to ensure accurate comparisons. Additionally, several key findings would benefit from appropriate statistical analyses to assess the significance of observed differences. Addressing these points would strengthen the authors' conclusions and further support the utility of Cas13 in RNA-targeting applications.

1. “Next, the *in vitro* collateral RNA cleavage activity of these proteins was assessed using the RNaseAlert® assay (IDT), which quantitatively detects non-specific RNA cleavage.”

I would appreciate clarification on how this reporter system distinguishes non-specific collateral activity from off-target binding by Cas13. Can it differentiate a single-nucleotide mismatch?

2. Is there a specific reason why the authors opted for RNP transfection rather than a plasmid-based method?

3. In Fig. 1a, “the total RNA of wild-type HAP1 cells transfected with LbuCas13a and a dEGFP-targeting guide RNA was intact at all timepoints.”

I assume all fluorescence intensities are normalized. However, the HAP1 samples show slightly elevated fluorescence signals at 1,000 nt and 4,000 nt even at early timepoints (e.g., 0, 20, and 50 minutes). Shouldn't the levels remain consistent across time if the RNA remains intact? What do these peaks represent? Also, the collateral activity is supposed to be non-targeted but is there any motif sequences for cleaved mRNA?

4. Related to point #3, a statistical comparison between the HAP1 and HAP1-dEGFP samples are needed.

5. For Fig. 1, what was the delivery efficiency of the RNP-based transfection? To accurately compare between cell lines, normalization based on delivery efficiency is necessary otherwise we could not eliminate the possibility that the difference

can be explained by the delivery efficiency of RNPs.

6. The authors state: "While target RNA cleavage seemingly started almost immediately, the collateral GAPDH cleavage started later and peaked at around 100 minutes after transfection. Both target and collateral RNA integrity returned to normal from about 500 minutes after transfection."

Could the authors elaborate on why collateral activity decreased after ~200 minutes? Does this correlate with the intracellular half-life of the Cas13b RNP complex?

7. "We electroporated HAP1 cells with LbuCas13a and a guide RNA targeting one of the highly abundant endogenous transcripts RPS19, GAPDH, or 18S rRNA."

This conclusion may not be fully supported by the data. RNA degradation was observed after delivery of LbuCas13a with targeting crRNA, but degradation could also result from indirect effects such as toxicity or apoptosis. Given that RPS19 and 18S rRNA are essential genes, their depletion could trigger cell death, complicating the attribution of observed degradation specifically to collateral activity.

8. "However, no collateral cleavage was observed when the LbuCas13a protein was expressed."

This statement is unclear. Was a crRNA delivered in this condition? If not, it would be helpful to clarify that this was a no-guide control.

9. In Fig. 2, the authors compare collateral activity across different cell lines, but delivery efficiency was not controlled. Since delivery efficiency likely varies across cell types, the comparison may not be accurate. A fluorescent reporter co-delivered with Cas13 (e.g., via 2A peptide or IRES) could enable sorting the transfected cells prior to RNA extraction.

10. Statistical tests are needed for Fig. 3B.

11. In Fig. 3B, baseline signal values at 0 hours should be shown for comparison.

12. In Fig. 4A, it was unclear what mixing ratio was used between HAP1 and dEGFP-HAP1 cells. Was it 1:4? Please clarify.

13. The authors explore the use of collateral activity to eliminate cells expressing a target mRNA. Could this strategy be adapted to target specific endogenous mRNAs — for instance, oncogenic transcripts — to selectively eliminate cancer cells from a heterogeneous population?

14. "The number of protein-coding transcripts decreased relative to all other transcript biotypes (Supplementary Fig. 11A)." Statistical testing should be provided.

15. "Due to LbuCas13a still being bound to the target RNA after purification, preventing reverse transcription at this location." Were RNA samples denatured before reverse transcription?

Version 1:

Reviewer comments:

Reviewer #1

(Remarks to the Author)

The authors have addressed all my comments.

Reviewer #3

(Remarks to the Author)

Bot et al. studied the mechanisms of Cas13-mediated collateral activities in mammalian cells. In the initial manuscript, there had been several points that need to be addressed, including statistical test, clarification of experimental setup. The revised manuscript has been improved significantly according to the revised figures and manuscript. The points have been reasonably addressed throughout the manuscript and I have no further comments on this manuscript.

Point to point response

We would like to thank the reviewers for their careful review of our manuscript. We're happy to say that we've been able to address all suggestions as outlined below.

We respond to all the reviewer suggestion in the table format. Modified figures are shown outside of the table directly below the relevant reviewer comment. Our modifications to the text are shown by blue highlighting in the manuscript document, and by Word's track changes function in this point by point response. At the end of the document we list additional changes we made to the manuscript not related to a specific reviewer comment.

The supplementary figures, supplementary note 1, and list of plasmid sequences can be found in the supplementary information file. The RNA-seq data has been deposited in the NCBI Gene Expression Omnibus (GSE220759). The total RNA-seq data with ERCC spike-in RNAs has been deposited at the NCBI Gene Expression Omnibus (GSE308909). The Nanopore sequencing data has been deposited at the NCBI Sequence Read Archive (PRJNA912090). Code used for the analysis of these sequencing experiments can be found on GitHub: <https://github.com/Geijsenlab/cas13>. All other source data is available in Supplementary Data 1 (main figures) and Supplementary Data 2 (supplementary figures).

Reviewer #1 (Remarks to the Author):

The study provides new insights into Cas13 activity in human cells, particularly its kinetics, cleavage preferences, and cellular responses. The Nanopore sequencing approach adds a mechanistic dimension, which could be useful for future Cas13 engineering. If rigorously validated, this could be useful for designing RNA-targeting CRISPR applications where collateral activity is harnessed (e.g., for cell elimination).

However, multiple claims require stronger validation, especially the unbiased nature of cleavage (uracil-containing loops) and transcriptome-wide RNA degradation. Is cell apoptosis truly due to Cas13 activity, or are other stress pathways involved? The claim of "major alterations to the total RNA profile" needs better quantification—does this include widespread transcript degradation or just a few highly expressed RNAs (e.g., GAPDH, RPS19, 18S rRNA)?

Also, control selection (e.g. RNA seq data) and experimental design issues may weaken their conclusions. Particularly, their RfxCas13d recombinant seems not functional, making comparisons among Cas13 orthologs potentially misleading.

Minor concerns:

Number	Reviewer comments	Reply
1	Suppl. Figure 10: The term transduction/empty transduction should be replaced with transfection for accuracy.	The term transduction has been replaced with transfection in what is now suppl. figure 11. Transduction has also been replaced with transfection in the supplementary figure legends of supplementary figures 3, 14 and 15.
2	Line 324: The term dEGFP-IRIS-PuroR appears to contain a typographical or conceptual error. Do the authors mean IRES (internal ribosome entry site) instead of IRIS? If so, this should be corrected for accuracy.	This is indeed a typographical error. IRIS has been replaced with IRES (now line 389). This error was also corrected in lines 501, 517, 518, 557, 724, and in supplementary file 1.

3	For the transcript coverage plots, why is the No iTOP condition (blue line) only shown at the 0-minute timepoint in Figure 6B, Figure 6C, and Supplementary Figure 12,13,14? To fully assess the impact of iTOP treatment over time, the No iTOP condition should also be included at later timepoints for proper comparison. If this data is unavailable, the authors should clarify why it was omitted.	The ‘no iTOP’ condition represents untreated cells, which were directly harvested in TRIZOL without being exposed to the iTOP transduction procedure. So they represent the before treatment timepoint, and should be read as the baseline coverage. We choose to plot them in the same column as the 0 minute after transfection samples for easier comparison with the other treatments, and because they were harvested simultaneously with these samples. We do not expect the transcript coverage of untreated cells to change in the 50 minutes to 24 hours experimental timeframe. However, to control for a possible decrease in RNA quality of the earlier timepoints due to (max 24h) longer storage in TRIZOL, the no iTOP samples were collected at the same time as the 0 minute after iTOP samples. What the transcript coverage seems to indicate (figure 6c and d, suppl. fig. 13-15), and the reviewer hints at with “assess the impact of the iTOP treatment over time”, is that the transcript coverage of the empty iTOP and non-targeting iTOP seems to drop a bit at 50 and 200 minutes after iTOP. In agreement with this, we also see a small reduction in the 28S rRNA peak height relative to the 18S rRNA at these timepoints (suppl. Fig. 11a). It is possible that the stress of the transfection causes some low level of background loss of RNA integrity in the cell. However, this effect is minor compared to the drop in transcript coverage due to activating LbuCas13a.
4	Lines 233 onwards: Ensure consistency in terminology—EGFP, dEGFP, or GFP. If these refer to distinct GFP variant sequences, please clarify; otherwise, standardize the naming throughout the text.	Occurrences of GFP in the section titled “Application of LbuCas13a as a cell selection tool” (from line 221) were a typographical error and have been changed to EGFP. Although we target dEGFP throughout most of the manuscript, in this section we indeed change to targeting fluorescent

		EGFP. The fluorescent EGFP variant enabled us to assess the proportion of EGFP positive cells by FACS after selection with Cas13. However, it is worth noting that the spacer sequence of the guide RNA is the same as when we are targeting dEGFP. We clarified this in the manuscript by inserting the following sentence on lines 226-229: “It is important to note that this EGFP targeting guide is the same guide RNA used to target dEGFP above, and does not cause collateral RNA cleavage or reduced viability in wildtype cells (Figure 1B1bd and Figure 3B3b)”
--	--	---

Major concerns:

Number	Reviewer comments	Reply
1	Suppl. Figure 1-RfxCas13d Collateral Activity: RfxCas13d did not show any collateral activity in this assay. Why? Cas13d is one of the most frequently reported orthologs exhibiting collateral activity. Control experiments are needed to determine whether the recombinant proteins are functional and whether the gRNAs (how many were used per ortholog?) are effective in achieving comparable target activity. To confirm on-target activity, in vitro cleavage assays against dEGFP and PAGE gel analysis of cleaved dEGFP RNA should be performed. Additionally, non-targeting control gRNAs are necessary to establish baseline activity.	We agree with reviewer comment 4 that the recombinant RfxCas13d protein we produced is likely inactive, and have therefore excluded most RfxCas13d data from the manuscript as suggested. We have made the following changes to the manuscript to clarify this:  - Added this text in line 91: RfxCas13d did not show any activity in this assay, nor did it show any activity using two other gRNAs (Supplementary Figure 1c), suggesting we did not manage to produce functional recombinant RfxCas13d protein. It was therefore excluded it from further testing. - Removed what was supplementary figure 3, which showed total RNA profiles after delivery of RfxCas13d with various guide RNAs in HAP1-dEGFP. - Removed what was supplementary figure 5c, which showed qPCR data of RfxCas13d experiments in HAP1-dEGFP. - Revised the paragraph in the discussion about the activity of different orthologs (lines 438-442).

		It now reads: While otherRecent reports have documented collateral cleavage or toxicity when using LwaCas13a^{1,2}, or RfxCas13d³⁻⁵. Unfortunately we were unable to produce active RfxCas13d protein. We did not find strong evidence of collateral RNA cleavage activity in human cells with these orthologs (see supplementaryby LwaCas13a, which may be due to differences in delivery method, cell line, target RNA or spacer sequence (Supplementary note 1). We have not been able to determine why our RfxCas13d protein was not functional. The isolated protein appears to be soluble and of the correct size (suppl. figure 1a). To see if the RfxCas13d plasmid perhaps acquired mutations during recombinant protein production we prepped the plasmid after induction and send it for whole plasmid sequencing. This did not reveal any mutations. On the other hand, the consistent low yield of our RfxCas13d purifications does suggest a problem with the production of RfxCas13d.
2a	Line 120 – Supplementary Figures 2 and 3 (Comparison of Orthologs to LbuCas13a in Human Cells): The comparison of different orthologs in Figure 1 and Supplementary Figures 2 and 3 is not entirely fair, as the delivery methods and controls differ between experiments.  • Figure 1A: HAP1-dEGFP data are overlaid with the negative control HAP1 cells. 	We indeed make use of both a negative control gRNA and WT HAP1 cells as negative control. Because dEGFP is an exogenous gene, it is possible to use WT cells that do not express GFP as a negative control. This gives extra information that a non-targeting gRNA would not, namely that the Cas13 activity we observe when using the dEGFP targeting guides is due to activation of Cas13 by the dEGFP target transcript, and not by off-target activation of Cas13 by some other endogenous transcript for example. Naturally, such a control is not possible when targeting endogenous housekeeping genes as we do in supplementary figures 2 and 3 of the original submission. For the RNA-seq and nanopore sequencing experiments, we choose to use a non-targeting gRNA as negative control instead of using wildtype cells. This was for two reasons. First, comparing two different cell

		lines (HAP1 and HAP1-dEGFP) would add an extra variable and thus lower the power of our experiment. Second, based on the results from figure 1 and the live imaging shown in figure 3 we were satisfied that the dEGFP targeting gRNA was specific.
2b	 Supplementary Figure 2: The overlay includes a non-targeting gRNA line with targeting gRNAs, but the specific cell line (HAP1-dEGFP or HAP1) is not indicated. 	These experiments were performed in HAP1-dEGFP. This information was added to the supplementary figure legend of this figure. For clarity, plot titles in the figure were changed from “LwaCas13a” to “Targeting with LwaCas13a in HAP1-dEGFP”. Name of line color legend changed from “Target” changed to “gRNA target”. Here a non-targeting gRNA was used, because beside dEGFP we are also targeting GAPDH and 18S rRNA.

Original version:

New version:

Number	Reviewer comments	Reply
2c	 Supplementary Figure 3: The authors used electroporation instead of iTOP (as used in Figure 1A and Supplementary Figure 2). Why was this approach chosen? Additionally, the control condition in Supplementary Figure 3 is unclear—there is no indication of HAP1-dEGFP or a non-targeting gRNA. 	We have followed your suggestion in point 4 to exclude data of RfxCas13d due to our recombinant protein likely being inactive, and have therefore removed supplementary figure 3. To still answer your questions in point 2c, despite multiple tries we were unable to

		produce sufficient recombinant RfxCas13d protein for iTOP delivery, which is why we choose to deliver the RNP by electroporation. To allow for a fair comparison with LbuCas13a, we performed electroporation delivery with RfxCas13d side-by-side with LbuCas13a under the same conditions. Other orthologs were not included in this comparison, as we already had selected LbuCas13a as the most promising among the other assessed orthologs. The used cell line was HAP1-dEGFP and a non-targeting gRNAs was used as control. Finally, we also used electroporation for the transfection of HEK293T (Suppl. Fig 6), because their plate attachment is too weak for iTOP delivery.
3	Lines 138–145 – Supplementary Figure 5 and Figure 1C: To ensure a valid comparison of ortholog activities, the authors should use consistent controls. Specifically: a. A non-targeting gRNA should be included in Figure 1C. b. Or, HAP1 cells should be included in Supplementary Figure 5.	We have added a non-targeting gRNA in what is now Figure 1d as per the reviewer’s suggestion. Three additional replicates of the dEGFP targeting condition in HAP1-dEGFP were included in this experiment. A 95% confidence interval has also been added to the regression lines based on a suggestion from another reviewer.

Original version:

New version:

d

Number	Reviewer comments	Reply
4	Lines 151–157, 387-388: The data do not support their conclusion that LbuCas13a exhibited the highest collateral RNA cleavage activity both in vitro and in cells. We lack strong evidence for collateral cleavage with these orthologs (they refer to RfxCas13d here, Ai et al., 2022; Y. Li et al., 2023; Shi et al., 2023). “...or RfxCas13d (Ai et al., 2022; Y. Li et al., 2023; Shi et al., 2023), we did not find collateral RNA cleavage activity in human cells with these orthologs.” The recombinant RfxCas13d they generated does not appear functional, which likely explains why their results contradict findings from other groups. To claim that a Cas13 variant lacks collateral activity, it must first be demonstrably active. Therefore, RfxCas13d should be excluded from the comparison after Figure 1, and any claims regarding its lack of collateral activity should be removed. Instead of arguing that LbuCas13a exhibits stronger collateral activity than RfxCas13d, they should focus on comparisons among functionally validated enzymes. Line 158 – Supplementary Note 1: The discussion in Supplementary Note 1 provides a more balanced interpretation of the data, acknowledging that the recombinant RfxCas13d may not have been functional in their in vitro assays. Their statement—“In any case, we cannot draw any conclusions about the in vitro activity of RfxCas13d.”—should	We agree with the reviewer that our recombinant RfxCas13d seems to be not functional. We have made the following changes to the manuscript to clarify this:  - Added this text in line 91: RfxCas13d did not show any activity in this assay, nor did it show any activity using two other gRNAs (Supplementary Figure 1c), suggesting we did not manage to produce functional recombinant RfxCas13d protein. It was therefore excluded it from further testing. - Removed what was supplementary figure 3, which showed total RNA profiles after delivery of RfxCas13d with various guide RNAs in HAP1-dEGFP. - Removed what was supplementary figure 5c, which showed qPCR data of RfxCas13d experiments in HAP1-dEGFP. - Revised the paragraph in the discussion about the activity of different orthologs (lines 438-442). It now reads: While otherRecent reports have documented collateral cleavage or toxicity when using LwaCas13a^{1,2}, or RfxCas13d³⁻⁵. Unfortunately we were unable to produce active RfxCas13d protein. We did not find strong evidence of collateral RNA cleavage activity in human cells with these orthologs (see supplementaryby LwaCas13a, which may be due to differences in delivery method, cell line, target

	be included in the main text (e.g., at Line 108, Suppl. Figure 1) to ensure transparency and prevent overinterpretation of the results.  • Revise the main text to remove misleading claims about RfxCas13d. • State that their recombinant RfxCas13d might not have been functional, preventing meaningful comparisons. • Incorporate key points from Suppl. Note 1 into the main text for clarity. 	RNA or spacer sequence (Supplementary note 1). We have not been able to determine why our RfxCas13d protein was not functional. The isolated protein appears to be soluble and of the correct size (suppl. figure 1a). To see if the RfxCas13d plasmid perhaps acquired mutations during recombinant protein production we prepped the plasmid after induction and send it for whole plasmid sequencing. This did not reveal any mutations. On the other hand, the consistent low yield of our RfxCas13d purifications does suggest a problem with the production of RfxCas13d.
5	Line 290, Figure 5A: What is the control used in this volcano plot? Please clarify which condition was used for normalization and specify the negative control.	Figure 5 has been reworked based on this comment and comment 7. Please refer to our response to comment 7 below for the detailed explanation. In short, we used DESeq2 with a design formula containing all variables of the experiment (protein + guide + targeting_RNP), and what we showed in the original volcano plot was the differential gene expression caused by the targeting variable.
6	Line 278: Before gaining insight into the cellular response to collateral RNA cleavage, the authors need to first assess the extent of global RNA degradation caused by collateral activity. To do so, they should include an earlier time point (50 min or 200 min) in their RNA-seq analysis. This would help determine whether widespread RNA degradation occurs shortly after Cas13 activation.	This is a great suggestion! We have now performed total RNA sequencing at 50, 200 and 1440 minutes after delivery (n=3). We used ERCC spike-in control RNAs for normalization between the samples. In short, this revealed that the loss of mRNAs is indeed near universal. It has also provided more insight into the temporal dynamics of the transcriptional response to collateral RNA cleavage. Altogether, this additional experiment has resulted into an entire new section being added to the results (which due to its length we do not show here, lines 338 to 370), a new main subfigure (Figure 6a) and a new supplementary figure (Suppl. Fig. 10). Due to total RNA sequencing being a second line of evidence next to the Nanopore sequencing, and it being much more high throughput than Nanopore sequencing, we are now more confident in our conclusion that collateral RNA cleavage causes widespread damage to cytoplasmic transcripts. We thank the reviewer for this suggestion.

7

Lines 288–290, 671–673: The authors claim "substantial changes in the transcriptome. Indeed, we found 721 significantly upregulated and 36 significantly downregulated genes in response to LbuCas13a activation." However, the Methods section states that four conditions were used:

- Empty transfection
- LbuCas13a without guide RNA
- LbuCas13a with a non-targeting guide RNA
- LbuCas13a with a dEGFP-targeting guide RNA (SP2)

Despite having three negative controls, Figure 5 presents only a single volcano plot and one GO analysis. It is unclear which control was used for normalization in these analyses. Was Cas13 + targeting gRNA normalized to an empty transfection, Cas13 alone, Cas13 + non-targeting gRNA? This must be explicitly stated in both the main text and the figure legend.

Additionally, the authors should provide volcano plots comparing Cas13 + dEGFP gRNA against the other two controls. Based on my experience with RNA-seq of CRISPR-expressing cells, the volcano plot in Figure 5 appears to be normalized against an empty transfection. If this is the case, the comparison is inappropriate, as overexpression of a bacterial protein alone can trigger an innate immune response—independent of Cas13 effector activity. Even a catalytically dead Cas13 can induce a similar effect. Without a clear statement on the chosen control for Figure 5 and without presenting plots for the other two negative controls, I am concerned that the authors have selectively presented their data.

To clarify, we used DESeq2 for the differential gene expression analysis, the code for this analysis is available on GitHub (<https://github.com/Geijsenlab/cas13>). We used a design formula with three variables: \sim protein + guide + targeting_RNP. The 'protein' variable indicates whether a sample was transduced with recombinant LbuCas13a protein. The 'guide' variable indicates whether samples were transfected with any gRNA (either non-targeting or targeting). Finally, the 'targeting' variable indicates whether samples were transfected with LbuCas13a and a dEGFP targeting gRNA:

	Protein	Guide	Targeting
Empty transfection	untrt	untrt	untrt
LbuCas13a only	trt	untrt	untrt
LbuCas13a with NT gRNA	trt	trt	untrt
LbuCas13a with a dEGFP-targeting gRNA	trt	trt	trt

As you probably know, DESeq2 will model how much of the variation in read counts can be explained by each variable and calculate log2 fold changes and p-values for each variable. Therefore, the volcano plot we showed in the original figure was relative to all control samples, as any differential expression due to for example an innate immune response to foreign protein or guide RNA is captured by the other variables in the design formula.

We agree that it is a good idea to show the volcano plots for the other variables as well, so they have now been added to figure 5 (a and b). The protein variable of the design formula only has 4 up and 12 downregulated genes, while the guide variable has just 1 downregulated gene. Perhaps the 16 hours between delivery and RNA isolation allowed the cells to resolve

most of their innate response to the foreign protein and RNA, or perhaps this cell line is more tolerant to foreign protein/RNA.

The following section has been added to the methods to better explain our DESeq2 analysis (lines 751-757):

A design formula was used with the variables 'protein', 'guide', and 'targeting'. The 'protein' variable indicates whether a sample had been transfected with recombinant LbuCas13a protein (all but the empty iTOP condition), the 'guide' variable indicates whether a sample had been transfected with a guide RNA (both the non-targeting and targeting condition), and the 'targeting' variable indicates whether a sample had been transfected with both LbuCas13a and a dEGFP targeting gRNA. For each variable of the design formula, genes with an adjusted P-value < 0.05 and log₂(Fold Change) > |1| were regarded as differentially expressed genes, and they were subjected to subsequent enrichment analysis.

The figure legend of figure 5 now reads: a, b and c Differential gene expression in response to transfections with LbuCas13a protein (a), guide RNA (b), or LbuCas13 protein and a targeting guide RNA (c). Fold change and P-values were calculated using DESeq2, see methods (n=3). ~~Dotted-Dashed~~ lines indicate adjusted P-value = 0.05 and |Log₂ FC| = 1.

A sentence in the results has been rewritten to better reflect the new figure (lines 316-319). It now reads:

Indeed, in response to LbuCas13a activation we found ~~721806~~ significantly upregulated and ~~3641~~ significantly downregulated genes (adjusted P-value < 0.05, |Log₂ FC| > 1) ~~in response), while the exposure to exogenous protein and/or RNA without LbuCas13a activation led to a only small number of differentially expressed genes~~ (Figure ~~5A5abc~~).

Finally, when remaking this figure based on this comment and the next, we re-ran the analysis with exactly the same code. In our previous analysis, the counts for the EGFP

transcript had erroneously not been included in the analysis. Furthermore, the DESeq2 package has been updated since then. Consequently, we found a slightly different number of differentially expressed genes in response to targeting than previously (806 upregulated instead of 721, and 41 downregulated instead of 36). Functional enrichment analysis has now been redone based on this re-analysis of the data, and revealed the same trends as previously. We have added a KEGG pathway enrichment analysis of the genes downregulated in response to dEGFP targeting to Suppl. Fig 9c and added a sentence in lines 323-324:

KEGG pathway enrichment analysis on the downregulated genes reveals enrichment of term related to fatty acid metabolism (Supplementary Figure 9c).

The KEGG plot of the upregulated genes is now shown in the main figure 5d. The GO biological process plot has been moved to supplementary figure 9d.

Original version:

New version:

Number	Reviewer comments	Reply
8	Figure 5A: Importantly, the authors need to indicate which dot represents dEGFP, the intended Cas13 activation and target. The volcano plot appears unrefined and requires improvement for publication. The figure legend resembles the default settings of EnhanceVolcano and should be revised to clearly define categories such as not significant, upregulated, and downregulated genes. The current labels (Log2FC, p-value, p-value and log2FC) are cutoffs and should be corrected for better interpretability. Additionally, the not significant gene group is missing from the plot, yet the legend includes 'Sig', which needs clarification. It is also unclear whether blue dots (in the legend) are present in	We agree, dEGFP is now indicated in the volcano plot of the response to dEGFP targeting. Although the fold change is negative (-0.83), this change was not significant ($p_{adj} = 1$). This is consistent with figure 1 and 6c, which show that the target RNA has mostly recovered by the timepoint of the RNA sequencing. The volcano plots in figure 5abc have been completely reformatted to enhance clarity. The EnhancedVolcano package is no longer used, instead plots are now made manually with ggplot2. Dots are now smaller. Colors are no longer used to indicate which dots pass the Log2FC and p-value thresholds, as this is already quite clear from the dashed lines indicating these threshold. Furthermore, the $-\log_{10}(p\text{-value})$ of some transcripts is so high that giving genes with non-significant p-value a different color just

	the plot. Lastly, given that the dataset contains 22,465 genes, the dot size is too large, reducing clarity. The size should be adjusted to improve readability and ensure most significant individual data points are distinguishable.	results in what appears as a horizontal line at 0, which is not very informative. The main message of these plots is that at 16 hours after iTOP transfection the cells have responded to the collateral RNA cleavage activity of LbuCas13a by upregulating immediate early stress response genes and immune response genes. The old and new versions of this plot are shown above in response to comment 7.
9	Lines 316–318: The statement "The abundance and read length of reads mapped to protein-coding transcripts greatly decreased at 50 and 200 minutes after transfection (Figure 6A and Supplementary Figure 11B)" is not fully supported by the data. In Figure 6A, the density plots appear lighter in color, and the center of the distribution shifts toward the bottom-right corner, indicating both a reduction in read abundance and a shortening of read length. However, in Supplementary Figure 11B, while the color intensity decreases (suggesting fewer reads), the center of the density plot does not shift. This implies a reduction in read abundance but no clear evidence of a change in read length. Additionally, the 0-minute timepoint data is missing in Supplementary Figure 11B, making it difficult to determine whether there was already a lower starting abundance. To strengthen the conclusions, the authors should provide the missing timepoint data and include at least one more biological replicate to confirm reproducibility.	We have performed an additional replicate of the Nanopore sequencing experiment as suggested. This new replicate had the highest collateral activity, while replicate 1 had the lowest (Suppl. fig. 11a). The density plot (Figure 6b) is now based on the reads of all three replicates combined. We figured this was the fairest way to show the data since the activity levels of Cas13 differed between replicates. In supplementary figure 12c we show the individual density plots of each replicate. There you can see that the population of transcripts that appears to stay intact at the 50 and 200 minutes timepoints in the main figure comes almost entirely from the first replicate, which had low Cas13 activity. Our conclusion that the read lengths are shorter at 50 and 200 minutes are now clearly supported by replicate 2 and 3. Replicate 1 also appears to show a population of shortened reads at the 200 minutes timepoint in supplementary figure 12c, and the smaller effect in replicate 1 can be explained by the low Cas13 activity of this replicate. In addition, in supplementary figure 11c we directly show the read length distributions of all samples and replicates. Reads at the 200 minutes timepoint are clearly shorter for all three replicates. We changed the y-axis of this plot to a log-scale to more clearly show this loss of longer reads. Finally, the global loss of protein coding transcripts is further supported by the additional total RNA-sequencing

experiment we performed on the reviewer's request (see comment 6).

Unfortunately, the 0 minute targeting timepoint of replicate 1 was lost due to failed library preparation, despite multiple attempts. Fortunately, the new replicate 3 is consistent with replicate 2. In short, we see no collateral cleavage on the density or coverage plots at 0 minutes after iTOP (figure 6c&d, suppl. fig 13). Furthermore, we see no degradation on the total RNA profiles at 0 minutes after iTOP (Figure 1a, Suppl. fig. 11a). Note that Suppl. Fig 11a includes the total RNA profile of the replicate 1 sample lost during library prep. Finally, 5'-3' qPCR shows target degradation, but no collateral cleavage at this timepoint (Figure 1d). Together, these data demonstrate good reproducibility across replicates and methods.

Additional change made:

A known artifact of Nanopore sequencing is that the second strand sometimes gets pulled into the pore immediately after the first strand, and gets added to the read of the first strand. This results in a population of reads that are 2x as long as the reference they map to. Such a population became apparent when we made the density plots of the new replicate 3 (here indicated by the blue arrow):

Old version density plot in the main figure (based on only replicate 2):

New version density plot fig. 6b (based on all three replicates):

The new density plots separated by replicate (suppl. Fig 12c):

Old version read length plot:

New version read length plot (suppl. Fig 11c):

Number	Reviewer comments	Reply
10 & 11	10: Lines 342–344: "Transcript coverage dips often map to predicted stem-loop structures, with uracil-containing single-stranded loops on the 5' side of the coverage dip, suggesting that the 5' side of the dip is the actual cleavage location". This statement lacks sufficient direct evidence. While the correlation between cleavage sites and predicted secondary structures is intriguing, the data do not conclusively demonstrate that the 5' side of the dip is the primary cleavage site. Additional validation, such as structural probing or mutagenesis experiments (on U), would be required to confirm this. Furthermore, transcript degradation and secondary structure mapping in high-throughput sequencing can be influenced by biases unrelated to direct Cas13 activity. The authors should provide more substantial experimental support before making this conclusion. 11: Line 350–352, 358: The statement "showing a preference for uracil-containing single-stranded loop structures" should be framed as a proposed model rather than a definitive conclusion. The current data suggest a correlation but do not establish causation. Additional experimental validation, such as structural probing or mutagenesis, would be necessary to confirm this preference.	Since comments 10 and 11 are similar we address them here together. It was indeed our intention to bring the apparent cleavage preference for uracil-containing single-stranded loop structures as a proposed model as suggested in comment 11, not as a definitive conclusion. Further experiments such as the proposed structural probing and mutagenesis are indeed very interesting suggestions, but would be an entire project itself and are outside the scope of this manuscript. We made the following changes to clarify that we are proposing a model and not definitive conclusions (Lines 406-410): In vitro, LbuCas13a was shown to prefer cleaving at uracil residues in or near single stranded loop structures⁶. This fits with our data, as transcript coverage dips often map to predicted stem-loop structures, with uracil containing single stranded loops on the 5' side of the coverage dip (Figure 7B7b), suggesting that the 5' side of the dip is the actual cleavage location preferences of LbuCas13a in cells is similar to its preferences in vitro. Lines 417-420: Additionally, it suggests that LbuCas13a cleaves RNA at specific positions, showing a preference for uracil-containing single stranded loop structures. We propose that the in vitro preference of LbuCas13a for cleaving at uracil-containing residues in single stranded loop structures is preserved in human cells. However, further experiments are

		needed to determine exact cleavage positions and structural preferences. We trust these revisions now clarify that the observed cleavage sites in the Nanopore sequencing data fit with previously reported in vitro preferences.
12	Line 349-350, Line 357: “LbuCas13a and suggests that 350 both mRNAs and cytoplasmic ncRNAs are cleaved without bias.” “All cytoplasmic RNAs seems to be cleaved in an unbiased manner.” This claim is not well supported by their data. Their evidence is based on reduced or shorter reads observed in nanopore sequencing, but they only closely analyze a few countable RNAs (RPS19, GAPDH, and 18S rRNA). To accurately assess the extent of Cas13-mediated RNA cleavage, the authors should include an earlier RNA-seq time point (e.g., 50 min or 200 min) to capture initial transcript downregulation and compile a comprehensive list of all cleaved transcripts aligned to the human reference genome. Additionally, the claim of unbiased cleavage overlooks spatial constraints on RNA accessibility. The cytoplasm contains structured, membrane-less organelles such as stress granules and P-bodies, which can sequester specific RNAs and limit Cas13 access. Certain ncRNAs also localize to membrane-bound organelles, further restricting their availability for cleavage. Without demonstrating that all cytoplasmic RNAs are equally accessible, the claim of unbiased cleavage is premature and should be revised to account for these biological constraints.	We realize that phrasing such as ‘without bias’ may suggest that all mRNAs are equally likely to be cleaved, which is indeed not entirely accurate. Therefore we have changed to wording of lines 413 to 417 to no longer imply equal cleavage probability for all mRNAs: Taken together, the total RNA sequencing with spike-in control RNAs and the Nanopore sequencing providesprovide further evidence of collateral RNA cleavage in human cells by LbuCas13a and suggests that both mRNAs and cytoplasmic ncRNAs are cleaved without biasnear universally subjected to collateral cleavage. The discussion has been rewritten and no longer contains any claims about the cleavage preference of LbuCas13a. We believe that with the additional data from the revision experiments, the conclusion of near global cytoplasmic RNA depletion is now much better supported. First, the total RNA-sequencing with ERCC spike-in RNAs provides a much higher number of reads than Nanopore sequencing. In short, it shows a near universal reduction of mRNAs, with the clear exception of mitochondrially encoded genes. As such, these mitochondrial transcripts show that RNAs within membraned compartments are protected and serve as an internal control as well. For more details see the new section of the manuscript (lines 338-370) and our response to comment 6. Second, another replicate of the nanopore sequencing has been performed, which further supports this conclusion, see our response to comment 9. Regarding differences in RNA accessibility due to spatial constraints, both the

		Nanopore and total RNA sequencing show that RNA within membrane-bound organelles (i.e. mitochondrial and nuclear RNA) is protected. Furthermore, the total RNA sequencing suggests that the sensitivity to cleavage of cytoplasmic mRNAs is correlated with their expression level. It is possible that highly expressed mRNAs are more sensitive due to being localized in more accessible parts of the cytoplasm. Interestingly, this trend has also been observed for the endogenous RNase RNASEL⁷. Finally, in the new section describing the total RNA-sequencing we describe the subset of mRNAs that are least affected by collateral cleavage or even increased in abundance. These turn out to be mostly stress response genes well known for their immediate upregulation in response to various stresses, suggesting their relative increase in abundance is due to increased transcription. Interestingly, a group of histone mRNAs are ‘upregulated’ at 200 minutes. It is possible these histone genes are upregulated as part of the stress response or due to altered cell cycle dynamics. Alternatively, histone mRNAs might be less sensitive to LbuCas13a due to their localization or distinct structure.
13	Line 390-393: This explanation is unclear. Are they suggesting that Cas13 degrades its own mRNA when expressed as a transgene, thereby reducing its protein levels? Or do they mean that transgene-expressed Cas13 is somehow less active than RNP-delivered Cas13? If guide RNA-independent cleavage is limiting expression, do they have data showing lower Cas13 protein levels in transgene-expressed cells compared to RNP delivery?	We have rewritten this section of the discussion (lines 444-456) and incorporated another relevant recent publication. It now reads: We found a clear correlation between the RNP concentration applied during delivery and the resulting collateral RNA cleavage. This suggests that the intracellular RNP concentration is an important factor, and that an efficient delivery method is important to generate collateral RNA cleavage. Recent reports show that besides the canonical target RNA activated RNase activity, Cas13 also exhibits both guide RNA independent and guide RNA dependent but target RNA independent RNase activity^{8,9}. This was shown to have a negative effect on cell viability, and thus may provide selection pressure against persistent Cas13 expression. Interestingly, recent reports of collateral cleavage by RfxCas13d in eukaryotic cell lines use transient plasmid

transfection^{3,4}, while publications finding no collateral cleavage by RfxCas13d use stable lentiviral integration^{10,11}. We hypothesize that these non-canonical RNase activities of Cas13 may limit the intracellular Cas13 contraction when Cas13 is expressed, which could also explain why we were unable to detect collateral cleavage when expressing LbuCas13a.

Unfortunately, assessing LbuCas13a protein levels after RNP delivery by iTOP has proven technically difficult. See reviewer 3 comment 9 for more details regarding this.

Reviewer #2 (Remarks to the Author):

This study by Bot et al. investigates whether CRISPR-Cas13 enzymes – which target RNA – exhibit the same collateral RNA cleavage in human cells as seen in bacteria and in vitro. The authors compare six Cas13 orthologs and find LbuCas13a uniquely capable of robust collateral RNA cutting in human cell lines when delivered as a ribonucleoprotein (RNP). Upon targeting a specific RNA (e.g., a GFP reporter), LbuCas13a rapidly cleaves not only the target but also other RNAs, causing a characteristic degradation pattern in total RNA within 50–100 minutes. This widespread RNA degradation triggers cellular stress responses and upregulation of innate immune genes, ultimately leading to apoptotic cell death by ~15–24 hours post-transfection. The team exploits this phenomenon as a cell elimination tool: cells expressing a target RNA can be selectively killed, allowing depletion of those cells from a mixed population. They demonstrate repeated rounds of Cas13a RNP delivery can progressively enrich or deplete cells based on expression of a target transcript. Finally, using Nanopore long-read sequencing, the authors map the precise temporal dynamics and cleavage sites of collateral damage. They show that LbuCas13a cleaves cytoplasmic mRNAs globally and without sequence bias (most transcripts are cut into short fragments), with a preference for cutting at uracil-rich loop structures in RNA, consistent with in vitro studies. Overall, the work provides a comprehensive picture of when, where, and how LbuCas13a causes collateral RNA cleavage in human cells, the resulting cellular responses, and a proof-of-concept application for selective cell ablation.

Collateral (bystander) RNA cleavage by Cas13 has been well-documented in bacteria and test tubes, but its occurrence in mammalian cells has been controversial and context-dependent. This study is novel in definitively showing that one particular Cas13 enzyme, LbuCas13a, can indeed trigger robust collateral RNA degradation in human cells under certain delivery conditions. Bot et al. not only identify LbuCas13a as an outlier with strong collateral activity, but also detail the kinetics (onset within <1 hour) and sequence preferences of cleavage inside cells – which to my knowledge has not been mapped at nucleotide resolution before.

In summary, the manuscript addresses a timely question in CRISPR biology, provides compelling evidence of a unique Cas13 behavior in human cells, and opens a new avenue for using CRISPR-Cas13 as a research and selection tool. These contributions are significant for the field of RNA-targeting CRISPR systems.

Please see detailed comments and suggestions below:

Number	Reviewer comments	Reply
1	The authors convincingly show LbuCas13a has strong collateral RNase activity in human cells when delivered as an RNP, but virtually no activity when expressed via plasmid. This discrepancy is important. A major point to address is why expression fails – the manuscript hints at guide-independent activity or insufficient protein levels. To bolster this, the authors could consider additional evidence or discussion. For instance, was any collateral RNA degradation detectable at later times or higher plasmid doses? Did the cells expressing Cas13a show any growth disadvantage (even if bulk RNA looked normal)? While not strictly necessary, exploring a lentiviral or inducible expression system might clarify if constant low Cas13a expression allows cells to adapt or if the enzyme is inherently self-limited. At minimum, expanding the discussion in Supplementary Note 1 (or main text) on why only RNP delivery works would be valuable. This will guide others in the field: it suggests that to observe collateral effects, transient high-level delivery is required, perhaps because cells otherwise quench the activity.	It is indeed interesting that we do not find strong collateral activity when transiently expressing LbuCas13a from a plasmid. We have also tried lentiviral delivery and interestingly could not even detect LbuCas13a on western blot, despite cells being resistant to the selection marker expressed on the same polycistronic mRNA as LbuCas13a (this negative result is not included in the manuscript). We have not noticed an obvious growth disadvantage for Cas13 transfected/infected cells. We have rewritten the section of the discussion about delivery (lines 444-456). It now reads: We found a clear correlation between the RNP concentration applied during delivery and the resulting collateral RNA cleavage. This suggests that the intracellular RNP concentration is an important factor, and that an efficient delivery method is important to generate collateral RNA cleavage. Recent reports show that besides the canonical target RNA activated RNase activity, Cas13 also exhibits both guide RNA independent and guide RNA dependent but target RNA independent RNase activity^{8,9}. This was shown to have a negative effect on cell viability, and thus may provide selection pressure against persistent Cas13 expression. Interestingly, recent reports of collateral cleavage by RfxCas13d in eukaryotic cell lines use transient plasmid transfection^{3,4}, while publications finding no collateral cleavage by RfxCas13d use stable lentiviral integration^{10,11}. We hypothesize that these non-canonical RNase activities of Cas13 may limit the intracellular Cas13 contraction when Cas13 is expressed, which could also explain why we were unable to detect collateral cleavage when expressing LbuCas13a.
2	The authors report collateral cleavage in all tested cell lines (HAP1, RD, U2OS, ARPE19) with no differences in the RNA fragment pattern (same fragments	Indeed, our main question here was to show qualitatively whether collateral RNA cleavage by LbuCas13a occurs in other cell lines besides HAP1. The quantitative

	appear). Interestingly, HAP1 had a quantitatively greater response. A major question is: what causes the difference in magnitude? The authors propose higher RNP uptake in HAP1 (since iTOP was optimized there). This is plausible, but could there be biological differences? For example, do some cells have higher RNase tolerance or RNA turnover that affects fragment detection? Or differences in innate immune priming that might degrade Cas13 sooner? While the current data suffice to say it's not cell-line specific qualitatively, the manuscript might acknowledge that efficiency can vary.	differences in activity between cell lines are certainly also interesting, and could be explained by many factors. We have expanded this section of the results to acknowledge this (lines 184-188): The HAP1-dEGFP cells we used in this study are mostlyindeed a mixture of haploid and diploid-A cells (Supplementary Figure 6b). It could be haploid cells are more sensitive to collateral activity due for example not being able to replace lost RNA as quickly as diploid cells. Another possible reason for the higher collateral activity in HAP1 cells could be a higher RNP delivery efficiency, as iTOP was developed using this cell type¹². lines 199-202: In conclusion, LbuCas13a exhibits collateral RNA cleavage in all tested cell lines. However, the activity varies between cell lines, which could be caused by differences in RNP delivery efficiency, cell ploidy, or other biological factors such as Cas13 protein and guide RNA degradation rates.
3	The study effectively demonstrates that LbuCas13a's ability to deplete target cells correlates with target RNA expression levels (Figure 4E), a crucial finding with significant implications for practical application of this cell selection system. However, this aspect warrants more detailed characterization to enhance the predictive power and utility of the approach. While the dCq values provide a relative measure of expression, establishing a quantitative minimum threshold of target RNA abundance required for effective cell elimination would make the findings more broadly applicable. Converting the relative measurements to absolute transcript copy numbers per cell (via digital PCR or RNA spike-in standards), determining at what expression level the depletion efficiency becomes suboptimal, and testing a broader panel of endogenous transcripts with varied abundance levels would provide researchers with clear guidelines for selecting appropriate target RNAs.	As part of this revision we have performed total RNA sequencing with ERCC spike-in RNA. This allowed us to quantify the exact concentrations of the used target RNAs in am/μl per 100 ng total RNA. The concentrations were 11.27, 19.36 and 254.12 am/μl for RPS19, GAPDH and dEGFP respectively. As you can see in the plot below, these are among the top of the expression levels (vertical dashed lines are the three targets mentioned above):  Unfortunately it is not really possible to determine an absolute minimum target RNA expression threshold. This is because collateral activity is also determined by other factors in addition to target RNA

		expression levels. First, the collateral activity is also highly dependent on the spacer sequence of the guide RNA. For example, in supplementary figure 8 we test five different CDK4 targeting guides and find one with high activity, two with about half that activity and two without any apparent activity. Second, it may depend on the delivery method. Third, activity levels between cell lines will probably differ. Some cell lines may be more sensitive to RNase activity than others, or have different intracellular Cas13 protein and guide RNA half-lives. Based on the new ERCC spike-in RNA sequencing data, we now mention in the discussion that all our used targets are in the top 1% of expression levels (lines 460): All target RNAs used in this study are in the top 1% of expression levels. We hope that this provides sufficient guidance for others to select their potential target RNAs.
4	The manuscript mentions that the HAP1 cells used are “mostly diploid,” despite this line’s description as near-haploid. Because ploidy status can significantly affect gene dosage, RNA expression levels, and susceptibility to CRISPR-based manipulations, it would strengthen the manuscript to thoroughly characterize the ploidy of the HAP1 population used in these experiments.	Thank you for this suggestion. We have assessed the ploidy of the HAP1-dEGFP line and found it to be a mixture of haploid and diploid cells (supplementary figure 6). To clarify this, we revised lines 184-187 to: The HAP1-dEGFP cells we used in this study are mostlyindeed a mixture of haploid and diploidA cells (Supplementary Figure 6b). It could be haploid cells are more sensitive to collateral activity due for example not being able to replace lost RNA as quickly as diploid cells.
5	The manuscript would benefit from a more detailed discussion of how the findings compare with previous studies showing limited collateral cleavage in eukaryotic cells.	As mentioned in our response to comment 1, we have expanded the discussion section about RNP delivery vs expression systems (lines 444-456). Here we now note that several studies showing limited collateral cleavage by RfxCas13d use lentiviral integration, while studies finding collateral cleavage use transient plasmid transfection. We have also revised the lines 438-442 of the discussion to: otherRecent reports have documented collateral cleavage or toxicity when using

		LwaCas13a^{1,2}, or RfxCas13d³⁻⁵. Unfortunately we were unable to produce active RfxCas13d protein. We did not find strong evidence of collateral RNA cleavage activity in human cells with these orthologs (see supplementary by LwaCas13a, which may be due to differences in delivery method, cell line, target RNA or spacer sequence (Supplementary note 1). The reason for the variability in collateral cleavage by Cas13 between different studies is a complex question. There are many variables that differ between studies, such as the target RNA expression level, spacer sequence, Cas13 ortholog used, delivery method, the specific cell line, and the methods used to assess collateral cleavage. All of these variables may also influence each other, making the matter even more complex. We have written a review about this (Bot et. al 2022) and refer to it in the discussion.
--	--	--

Reviewer #3 (Remarks to the Author):

Bot et al. investigated the mechanisms of Cas13-mediated collateral activity across multiple mammalian cell lines and proposed its potential application for targeted cell elimination. The application would be of high potential if the method could be used for endogenous transcripts, such as oncogenes. The study is generally well designed, and the data provide valuable insights into the temporal kinetics of Cas13-induced RNA degradation. However, some measurements obtained from different cell lines or samples need to be normalized based on delivery efficiency to ensure accurate comparisons. Additionally, several key findings would benefit from appropriate statistical analyses to assess the significance of observed differences. Addressing these points would strengthen the authors' conclusions and further support the utility of Cas13 in RNA-targeting applications.

Number	Reviewer comments	Reply
1	“Next, the in vitro collateral RNA cleavage activity of these proteins was assessed using the RNaseAlert[®] assay (IDT), which quantitatively detects non-specific RNA cleavage.” I would appreciate clarification on how this reporter system distinguishes non-specific collateral activity from off-target binding by Cas13. Can it differentiate a single-nucleotide mismatch?	The IDT RNaseAlert reporter consists of a fluorophore and a quencher tethered together by a short RNA sequence (see image below). RNase activity results in cleavage of this RNA linker, liberating the fluorophore from the quencher. For clarity, in this assay the RNA linker of the RNaseAlert reporter is not the target RNA of Cas13. After Cas13 is activated by the target RNA the RNaseAlert reporter is cleaved collaterally. Supplementary figure 1b shows that the rate of fluorescence increase is positively correlated with the target RNA concentration,

		and is close to 0 for most orthologs at the lowest target RNA concentration, illustrating that the fluorescent signal of the RNaseAlert assay depends on activation of Cas13 by a target RNA. It is true that different Cas13 orthologs have different cleavage preferences. IDT does not share the sequence of the RNaseAlert RNA linker. So it is indeed possible that some Cas13 orthologs have a higher affinity for cleaving it than others. We mention this in the discussion (lines 434-436): However, different orthologs may have different preferences for factors such as the assay buffer, the protospacer sequence^{72,76} and the cleavage substrate²¹. 2	Is there a specific reason why the authors opted for RNP transfection rather than a plasmid-based method?	Our lab is focused on RNP delivery and the iTOP RNP delivery method has been developed in our lab. RNP delivery has a number of advantages. It provides flexibility. Changing the guide RNA is simple, and doesn't require genetic manipulation of the cells. The RNP concentration can easily be titrated and can potentially reach higher intracellular concentrations than expression systems. In addition, it may decrease any potential unintended toxic or selection effects, as RNP delivery is transient. For example, for Cas13 toxicity due to constitutive non-canonical RNase activity has been reported^{8,9}.
3	In Fig. 1a, “the total RNA of wild-type HAP1 cells transfected with LbuCas13a and a dEGFP-targeting guide RNA was intact at all timepoints.” I assume all fluorescence intensities are normalized. However, the HAP1 samples show slightly elevated fluorescence signals at 1,000 nt and 4,000 nt even at early timepoints	The peaks just after 1000 and at ~4000nt are the 18S and 28S ribosomal RNA peaks. Ribosomal RNAs are longer than that (1.9 and 5 kb respectively), however on the bioanalyzer they migrate faster than would be expected from their size, probably due to their structure. The figure legend erroneously said all data were normalized to the RNA concentration, however only the bottom row with the zoomed in plots are normalized. This is now fixed in both the legend and y axis label. We left the top row

(e.g., 0, 20, and 50 minutes). Shouldn't the levels remain consistent across time if the RNA remains intact? What do these peaks represent?

Also, the collateral activity is supposed to be no-targeted but is there any motif sequences for cleaved mRNA?

unnormalized so that you can see the RNA intensities decrease over time for the HAP1-dEGFP targeted cells. However, there indeed also seems to be an effect on HAP1, leading to lower peaks at 100 and 200 minutes with a bigger effect on the 28S rRNA peak. Therefore, it is indeed not accurate to claim a complete lack of RNA degradation in the control conditions. We have changed this sentence to only mention the Cas13 specific degradation fragments (lines 100-104):

The total RNA profile of dEGFP expressing cells transfected with LbuCas13a and a dEGFP targeting guide RNA clearly revealed a degradation pattern mainly consisting of several novel fragments preceding the 18S ribosomal RNA (rRNA) peak (Figure 1A), ~~while 1a~~. ~~These fragments did not appear in~~ the total RNA of wildtype HAP1 cells transfected with LbuCas13a and a dEGFP targeting guide RNA ~~was intact at all timepoints~~.

Regarding sequence motifs for RNA cleavage by Cas13, they generally have a preference for cleavage next to a specific nucleotide. For example, LbuCas13a has a known preference for cleaving uracil. Furthermore, Cas13 prefers to cleave loop structures of stem loops. Our Nanopore experiment suggests these known *in vitro* preferences are maintained in cells (Figure 7).

Old version:

New version:

Number	Reviewer comments	Reply
4	Related to point #3, a statistical comparison between the HAP1 and HAP1-dEGFP samples are needed.	We have added statistics to compare the HAP1 and HAP1-dEGFP sample (Figure 1b). We opted to calculate the area under the curve where degradation fragment occur and compare those values between the two cell lines, instead of compare the raw bioanalyzer data directly. This was because the bioanalyzer measures the fluorescence many times per second, and the raw data thus contains what could be considered many repeated measurements. In addition, in figure 1d we have added 95% confidence interval bands. We choose to display the confidence interval, because three treatment are shown per timepoint and it would thus be difficult to clearly show which was compared with which if we showed p-values. A condition with a non-targeting guide in HAP1-dEGFP and three additional replicates of with the dEGFP targeting guide in HAP1-dEGFP have been added based on a suggestion from another reviewer.

New subfigure (1b):

Original version (fig 1d, was 1c):

New version (Fig 1d):

Number	Reviewer comments	Reply
5	For Fig. 1, what was the delivery efficiency of the RNP-based transfection? To accurately compare between cell lines, normalization based on delivery efficiency is	We have added a transfection of HAP1-dEGFP with LbuCas13a and a non-targeting guide to figure 1d, showing that in the exact same cell line collateral cleavage only happens if the guide has a matching target. For the total RNA profiles

	necessary otherwise we could not eliminate the possibility that the difference can be explained by the delivery efficiency of RNPs.	these data were already present in supplementary figure 11a. As discussed below at comment 9, we have so far not been able to accurately assess intracellular Cas13 protein levels after iTOP. However, the iTOP transfection procedure was developed in our lab using Cas9 and the HAP1 cell line¹², so we are very confident that HAP1 can efficiently be transfected using iTOP. We still routinely use the wildtype HAP1 line for gene editing experiments with Cas9 or Cas12 and reach high efficiencies using iTOP. Furthermore, the HAP1-dEGFP line is a polyclonal line directly derived from wildtype HAP1 by lentiviral infection with the dEGFP construct, so big differences in delivery efficiency between these two lines are not expected.
6	The authors state: “While target RNA cleavage seemingly started almost immediately, the collateral GAPDH cleavage started later and peaked at around 100 minutes after transfection. Both target and collateral RNA integrity returned to normal from about 500 minutes after transfection.” Could the authors elaborate on why collateral activity decreased after ~200 minutes? Does this correlate with the intracellular half-life of the Cas13b RNP complex?	We think that this is indeed because Cas13 protein and guide RNA get degraded over time. For Cas9 it is well known that most protein is removed within 24h after RNP delivery (see for example¹³). We expect that the gRNA has an even shorter half-live, especially since we here used guide made by in vitro transcription without any protecting RNA modifications. We have tried to assess the intracellular protein concentrations after iTOP delivery, however we have not been successful. See our response to comment 9 for a more detailed explanation. Because we unfortunately have no data in this regard, we do not want to speculate in the manuscript about the cause of the reduction in collateral activity after 200 minutes.
7	“We electroporated HAP1 cells with LbuCas13a and a guide RNA targeting one of the highly abundant endogenous transcripts RPS19, GAPDH, or 18S rRNA.” This conclusion may not be fully supported by the data. RNA degradation was observed after delivery of LbuCas13a with targeting crRNA, but degradation could also result from indirect effects such as toxicity or apoptosis. Given that RPS19 and 18S rRNA are essential genes, their depletion could trigger cell death, complicating the attribution of	It is indeed difficult to disentangle what downstream effects are due to specific target RNA knockdown and what are due to collateral cleavage when targeting an endogenous transcript. This is why throughout most of the manuscript we target the exogenous dEGFP. However, we believe that it is very unlikely that a target RNA specific effect contributes to the observed total RNA degradation pattern in figure 2.  - First, the degradation pattern we observe when targeting the endogenous GAPDH, RPS19 and 18S rRNA is exactly the same as when we target the exogenous dEGFP. We are aware of only one endogenous system that generates

	observed degradation specifically to collateral activity.	a total RNA degradation pattern similar to LbuCas13a, namely RNASEL. However, the RNASEL system is specifically triggered by viral dsRNA. As part of another as of yet unpublished project we have data showing RNASEL is not triggered by LbuCas13a collateral cleavage (we see no reduction in the total RNA degradation pattern after targeting with LbuCas13a in cells treated with an RNASEL inhibitor or in RNASEL KO cells).  - Second, we harvested RNA just 100 minutes after delivery. The proteins of the housekeeping genes we target have very long half-lives, namely >30h¹⁴ and several days¹⁵ for GAPDH and RPS19 respectively. Therefore, it is unlikely a significant reduction in the levels of these proteins has occurred within this timeframe, and that cells have been able to detect this and respond by activating some endogenous RNase. - Third, in figure 3 we show that the initial stages of apoptosis start much later, at ~7 hours after iTOP.
8	“However, no collateral cleavage was observed when the LbuCas13a protein was expressed.” This statement is unclear. Was a crRNA delivered in this condition? If not, it would be helpful to clarify that this was a no-guide control.	This sentence summarizes the results of this paragraph, in which we indicate that guide RNAs were delivered into LbuCas13a expressing cells (lines 156-160): Both the unsorted and enriched cells were transfected with a dEGFP-targeting guide RNA after allowing the enriched cells to recover overnight from sorting. No collateral cleavage pattern was observed in the protein expressing cells transfected with guide RNA (Supplementary figure 7B5c)
9	In Fig. 2, the authors compare collateral activity across different cell lines, but delivery efficiency was not controlled. Since delivery efficiency likely varies across cell types, the comparison may not be accurate. A fluorescent reporter co-delivered with Cas13 (e.g., via 2A peptide or IRES) could enable sorting the transfected cells prior to RNA extraction.	Our main objective for figure 2 was to show qualitatively that collateral cleavage by LbuCas13a was not limited to the HAP1 cell line. We show that collateral activity occurs in all tested cell lines. Indeed the activity varies between cell lines, and it is very likely delivery efficiency is one of the reasons. In addition, there are probably other biological differences between lines that play a role, such as the ploidy of the cells. The results now specifically mention these possible reasons for the difference in activity between cell lines (lines 184-188): The HAP1-dEGFP cells we used in this study are mostly indeed a mixture of haploid and diploid. A

		cells (Supplementary Figure 6b). It could be haploid cells are more sensitive to collateral activity due for example not being able to replace lost RNA as quickly as diploid cells. Another possible reason for the higher collateral activity in HAP1 cells could be a higher RNP delivery efficiency, as iTOP was developed using this cell type¹². We have tried various things to assess the protein concentration after iTOP inside the cells. Unfortunately, coupled expression systems like 2A or IRES cannot be used, because we are directly delivering Cas13 protein into the cells. The protein concentration used in the iTOP transfection mix is really high (15 μM). A consequence of this high concentration is that a lot of protein stays behind in the well outside of the cells after the transfection is complete, including a lot seemingly sticking to the outside of the cell membrane. Even with several washes, we have found it impossible to accurately distinguish between internal and external protein using microscopy. We have also not been able to set up a western blot assay for the same reasons. When we directly harvest cells we see a prominent band on western blot, however most of this protein is most likely leftover extracellular protein. When we first extensively treat the cells with trypsin before harvesting in order to digest this extracellular protein, surprisingly we see no band on western blot at all. Since we clearly show high collateral activity throughout this paper we believe there must be a technical reason why we do not see LbuCas13a protein on western blot when first using trypsin to remove leftover extracellular protein. Despite extensive attempts, we have unfortunately not been able to identify the cause of this issue.
10	Statistical tests are needed for Fig. 3B.	Thank you for this suggestion. We have performed a Repeated-measure ANOVA to compare number of Annexin V and CellTox Green positive cells between the cell lines and guides used in figure 3b. We added the following to the figure legend of figure 3B: The number of Annexin V positive cells varied significantly between cell lines (Repeated-measures ANOVA, $F_{1,15} = 9.565$, $p = 0.00743$) and treatments (Repeated-measures ANOVA, $F_{3,15} = 4.812$, $p = 0.01530$). The number of CellTox Green positive cells was not significantly

		different between cell lines (Repeated-measures ANOVA, $F_{1,15} = 1.093$, $p = 0.3123$) and treatments (Repeated-measures ANOVA, $F_{3,15} = 1.225$, $p = 0.3350$).
11	In Fig. 3B, baseline signal values at 0 hours should be shown for comparison.	Imaging was only started at 5 hours after iTOP. This was necessary to allow the cells to recover from the iTOP procedure, and to allow for the cells and plate to equilibrate to the microscope chamber before starting imaging. Otherwise cells would move out of focus during the time course. At 5 hours after iTOP the absolute number of Annexin V and CellTox Green positive cells is still very similar for all conditions, showing that cell death induced by collateral cleavage had not started yet.
12	In Fig. 4A, it was unclear what mixing ratio was used between HAP1 and dEGFP-HAP1 cells. Was it 1:4? Please clarify.	We aimed for about 75% EGFP positive cells when mixing the cells. This information has now been added to the legend of figure 4.
13	The authors explore the use of collateral activity to eliminate cells expressing a target mRNA. Could this strategy be adapted to target specific endogenous mRNAs — for instance, oncogenic transcripts — to selectively eliminate cancer cells from a heterogeneous population?	This indeed expand the relevance of our work, thank you for the suggestion. As part of the revision we have conducted extensive experiments targeting the endogenous CDK4, which is overexpressed in the rhabdomyosarcoma line RH30 due to a genomic amplification. As a control we used the rhabdomyosarcoma line RD, which does not have this genomic amplification. These experiments have resulted in a new subfigure (4f), new supplementary figure (suppl. fig. 8), a new section being added to the results (lines 266-299, not shown here due to its length), and to the discussion (lines 461-469, not shown here due to its length). In short, CDK4 expression is about 14x higher in RH30 than in RD (Suppl. Fig. 8a). We compared the activities of 5 different CDK4 targeting guide RNAs and show that collateral cleavage is induced in RH30 by some of these guides, but not in RD (Suppl. Fig 8bc). We next conduct a titration of the protein and guide concentration (Supl. fig. 8de). We performed a cell viability assay showing CDK4 targeting results in a significantly bigger decrease in viability in RH30 than in RD (Supl. fig. 8f). Finally, as a proof of principle we mixed RH30 and RD cells and show that Cas13 can be used to select against RH30 by targeting CDK4 (figure 4f).
14	“The number of protein-coding transcripts decreased relative to all	Because in the original submission we had just 2 replicates of the nanopore sequencing, statistics

	other transcript biotypes (Supplementary Fig. 11A).” Statistical testing should be provided.	were not possible. With the addition of the third replicate, we found that there were no statistically significant differences between the fraction of reads mapping to different transcript biotypes. We changed lines 379-381 to: We Consistent with the total RNA sequencing above, we found that when targeting dEGFP, the number of protein coding transcripts decreased relative to all other transcripts biotypes when targeting dEGFP, although this difference was not significant (Supplementary figure 11A). On the other hand, Figure 12b. The reason for the decrease in the proportion of reads mapping to protein coding transcripts not being significant is probably the huge variation in the number of snRNA reads. snRNAs can account for anything from ~10% to ~50% of the reads. We believe this is a technical artifact. Several bead cleanups are part of the library prep protocol. We aimed to use a bead ratio such that fragments >100 bp are kept. This is right around the size of snRNA, meaning that small variation in the ratio between the volume of beads and sample may result in snRNAs being maintained in some samples but not others. Several lines of evidence still support loss of mRNAs due to collateral RNA cleavage by LbuCas13a. The new total RNA sequencing experiment with spike-in RNAs show a near global depletion of mRNAs, while mitochondrial and nuclear transcripts were unaffected (Figure 6a and supplementary figure 10). The nanopore density plots of the protein coding transcripts show both a reduction in size of the fragments and a decrease in their abundance (Figure 6b). The Nanopore transcript coverage plot also show a consistent decrease in coverage for mRNAs, but not for mitochondrial or nuclear transcripts (Figure 6cd, Supplementary figures 13-15).
15	“Due to LbuCas13a still being bound to the target RNA after purification, preventing reverse transcription at this location.” Were RNA samples denatured before reverse transcription?	This is a good point. The rRNA depletion contains a 10 minutes 70°C denaturing step, and the reverse transcription contains a 5 minute 65°C denaturing step before the RT enzyme is added. It is true that it is unlikely that LbuCas13a would survive RNA isolation and these denaturing steps to block the reverse transcriptase. However, Cas13a performing guide-directed cleavage at the exact protospacer location goes against all established literature, so we felt we should at

least mention this alternative hypothesis even though it is unlikely.

Other modifications to the manuscript

- The abstract has been rewritten to incorporate the new experiments and reduce the size to the recommended 150 words. It now reads:

CRISPR-Cas13 exclusively targets RNA. In ~~prokaryotic cells~~ prokaryotes, Cas13 cleaves both target and non-target RNA indiscriminately upon activation by a specific target RNA, but in eukaryotic cells collateral cleavage activity has been limited. ~~To investigate collateral cleavage by Cas13 in eukaryotic cells, we first compared various Cas13 orthologs and found that specifically~~ Here we report that LbuCas13a exhibits strong collateral RNA cleavage activity in human cells when delivered as ribonucleoprotein, independent of cell line and targeting both exogenous and endogenous transcripts. Collateral RNA cleavage ~~started~~ starts within 50 minutes of ribonucleoprotein delivery resulting in major alterations to the total RNA profile. In response to the collateral RNA cleavage, cells upregulated genes associated with the stress and innate immune response, ultimately leading to apoptotic cell death. This enabled us to use LbuCas13a as a flexible and repeatable target-RNA-specific cell elimination tool. Finally, ~~we used~~ using both total RNA sequencing and Nanopore sequencing ~~to explore the identity of collaterally cleaved RNAs, the nucleotide position at which they are cleaved, and the temporal dynamics of collateral RNA cleavage. This revealed, we found~~ that LbuCas13a activation leads to rapid and near global cleavage ~~depletion~~ of cytoplasmic RNAs, and that cleavage occurs at specific nucleotide positions. ~~In conclusion, we here report that LbuCas13a has high collateral activity in human cells and describe the temporal dynamics of the collateral RNA cleavage, the cellular responses ultimately leading to apoptosis, how this can be exploited as a cell elimination tool, and the collateral cleavage preferences of LbuCas13a.~~

- The individual data points have been added to the plots of figure 2c and suppl. Fig. 1b.

Old version 2c:

New version 2c:

c

Old version Suppl. Fig 1b:

B

New version Suppl. Fig 2b:

b

- Error in the sentence starting in introduction line 61: ~~In this~~We set out to explore the on-target and collateral RNA cleavage activity of LbuCas13a in cells.
- Expanded lines 70-74 of the introduction: We demonstrate that this can be used to enrich for cells that have undergone gene editing, and select against cancer cells by targeting an overexpressed oncogene. Finally, using both total RNA sequencing with ERCC spike-in RNAs and Nanopore sequencing, we observed that upon activation by the target RNA, LbuCas13a indiscriminately cleaves collateral RNA cleavage leads to near global depletion of cytoplasmic RNAs.

- ~~Modified sentence starting at line 127: Both target and c~~Collateral RNA integrity returned to normal from about 500 minutes after transfection.
- Modified sentence starting at line 131: ~~Although~~ LwaCas13a, BzCas13b and PspCas13b showed a minimal increase in non-target RNA 5' Cq values compared to the non-targeting control (Supplementary Figure 4b), suggesting low levels of collateral cleavage, however these increases were not significant.
- Made small changes to sentence starting at line 139: ~~While collateral~~Collateral cleavage inside cells has also been reported for some of the other orthologs compared here³⁻⁵, however we could not find strong evidence for collateral cleavage with these orthologs.
- Rewrote sentence at lines 237-239 from: We performed ~~several~~three serial transfections at 2-day intervals, ~~with up to 3 subsequent selection rounds~~.
- At line 373, inserted one word: We decided to use Nanopore sequencing to further explore the identity of the RNAs that are subject to collateral cleavage, ...
- Changed figure 4c to show the multiple EGFP targeting round by line type instead of color, and to show individual data points:

Old version:

New version:

- The discussion has been rewritten to incorporate revision experiments, reviewer suggestions, and shorten it by removing repetitions of conclusions etc. (not shown here due to length).
- In supplementary figure 11a, the bioanalyzer traces of the non-targeting treatments of replicate two were slightly misaligned, which has now been fixed (left = old, right = new):

- In Suppl. Fig. 2 various occurrences of “18S” have been changed to “18S rRNA” for clarity
- All letters indicating figure panels were changed to lowercase bold
- Added a Statistical analysis section to the methods
- Added a Data availability section to the methods
- Added a Code availability section to the methods

References

1. Wang, Q. *et al.* The CRISPR-Cas13a Gene-Editing System Induces Collateral Cleavage of RNA in Glioma Cells. *Advanced Science* 1901299, (2019).
2. Özcan, A. *et al.* Programmable RNA targeting with the single-protein CRISPR effector Cas7-11. *Nature* 597, (2021).
3. Shi, P. *et al.* Collateral activity of the CRISPR/RfxCas13d system in human cells. *Commun Biol* 6, (2023).
4. Li, Y. *et al.* The collateral activity of RfxCas13d can induce lethality in a RfxCas13d knock-in mouse model. *Genome Biol* 24, 1–25 (2023).
5. Ai, Y., Liang, D. & Wilusz, J. E. CRISPR/Cas13 effectors have differing extents of off-target effects that limit their utility in eukaryotic cells. *Nucleic Acids Res* 50, 1–16 (2022).
6. East-Seletsky, A. *et al.* RNA Targeting by Functionally Orthogonal Type VI-A CRISPR-Cas Enzymes. *Mol Cell* 66, 373–383 (2017).
7. Burke, J. M., Moon, S. L., Matheny, T. & Parker, R. RNase L Reprograms Translation by Widespread mRNA Turnover Escaped by Antiviral mRNAs. *Mol Cell* 75, 1203-1217.e5 (2019).
8. Li, Z. *et al.* Intrinsic targeting of host RNA by Cas13 constrains its utility. *Nat Biomed Eng* 8, 177–192 (2023).
9. Liu, W. *et al.* RNA target-independent non-canonical activation (RINCA) of Cas13 trans-nuclease activity. *Sci Bull (Beijing)* <https://doi.org/10.1016/j.scib.2025.07.015> (2025) doi:10.1016/j.scib.2025.07.015.

10. Wessels, H. H. *et al.* Prediction of on-target and off-target activity of CRISPR–Cas13d guide RNAs using deep learning. *Nat Biotechnol* 42, 628–637 (2024).
11. Tieu, V. *et al.* A versatile CRISPR–Cas13d platform for multiplexed transcriptomic regulation and metabolic engineering in primary human T cells. *Cell* 187, 1278–1295 (2024).
12. D’Astolfo, D. S. *et al.* Efficient intracellular delivery of native proteins. *Cell* 161, 674–690 (2015).
13. Kim, S., Kim, D., Cho, S. W., Kim, J. J.-S. J. & Kim, J. J.-S. J. Highly efficient RNA-guide genome editing in human cells via delivery of purified Cas9 ribonucleoproteins. *Genome Res* 128, 1–32 (2014).
14. Dhaliwal, N. K., Abatti, L. E. & Mitchell, J. A. KLF4 protein stability regulated by interaction with pluripotency transcription factors overrides transcriptional control. *Genes Dev* 33, 1069–1082 (2019).
15. Miyake, K. *et al.* Development of cellular models for ribosomal protein S19 (RPS19)-deficient diamond-blackfan anemia using inducible expression of siRNA against RPS19. *Molecular Therapy* 11, 627–637 (2005).

The reviewers did not have any additional comments:

Reviewer #1 (Remarks to the Author):

The authors have addressed all my comments.

Reviewer #3 (Remarks to the Author):

Bot et al. studied the mechanisms of Cas13-mediated collateral activities in mammalian cells. In the initial manuscript, there had been several points that need to be addressed, including statistical test, clarification of experimental setup. The revised manuscript has been improved significantly according to the revised figures and manuscript. The points have been reasonably addressed throughout the manuscript and I have no further comments on this manuscript.